# Research on the coupling relationship and interaction between urbanization and eco-environment in urban agglomerations: A case study of the Chengdu-Chongqing urban agglomeration

**Weilong Wu[1,2], Ying Huang[1], Yuzhou Zhang[2], Bo Zhou[1] ***

**1** School of Architecture and Environment, Sichuan University, Chengdu, Sichuan, Chin, **2** Hubei Key Laboratory of Biological Resources Protection and Utilization, Hubei Minzu University, Enshi, Hubei, China

* zxt001@163.com

**Data Availability Statement:** All relevant data are within the manuscript and its Supporting Information files.

## Abstract

Urban agglomerations are emerging as new regional units for national participation in global competition and the international division of labor. However, they face increasingly severe resource and eco-environment pressures during urbanization. The coordination of the relationship between urbanization and the eco-environment has attracted global attention. In this study, we used Coupling Coordination Degree and Vector Autoregression models to examine the dynamic evolution, coupling relationships, coordinated development patterns, and interaction mechanisms between urbanization and the eco-environment. The results indicate that: (1) The level of urbanization in the Chengdu-Chongqing Urban agglomeration was relatively low, and the region showed a good eco-environment background. However, rapid urbanization is gradually straining the carrying capacity of the eco-environment. (2) A close and stable coupling relationship exists between urbanization and the eco-environment, which has reached an advanced coupling stage. The status of coordinated development among cities differs considerably, and multiple stable forms may exist simultaneously. (3) Urbanization has a substantial impact on environmental changes, whereas the restrictive effect of the eco-environment on urbanization development is not particularly notable. (4) Various interactive relationships exist between the urbanization and eco-environment subsystems, including positive promotion and negative constraint effects. The positive promotion effect mainly manifests between the economic, social, and ecological response subsystems, while the negative constraint effect is most evident in the mutual coercion and inhibition between the regional urbanization, economic urbanization, ecological status, and ecological pressure subsystems. These findings have important policy implications for decision makers exploring the path of coordinated and sustainable development in urbanization and the eco-environment in Urban agglomerations.

**Funding:** This work was supported by the Key Research and Development Program of Sichuan Province (2023YFS0368). There was no additional external funding received for this study.

**Competing interests:** The authors have declared that no competing interests exist.

# 1 Introduction

The coupling relationship and interaction between urbanization and the eco-environment are extremely important aspects in the study of human-environment relationships. It is one of the core contents of modern human geography and physical geography research, and it is also an important topic for achieving the United Nations' Sustainable Development Goals by 2030 [1, 2]. With the advent of the Anthropocene, understanding the interaction between humans and nature in the process of urbanization is crucial for pursuing human well-being and global sustainable development.The main carrier of urbanization, urban agglomerations are gradually becoming a new territorial unit for national participation in global competition and the international division of labor; they are of great significance for national and regional economic development, international competitiveness, and sustainable development [3]. However, the rapid urbanization and industrialization within Urban agglomerations have resulted in increasingly severe resource and environmental pressures, posing numerous sustainable development challenges for cities [4], including air pollution [5], traffic congestion [6], energy shortage [7], water quality deterioration [8], ecosystem degradation [9], forest/vegetation coverage and the loss or decline in biodiversity [10, 11].These ultimately lead to a decline in the quality of urbanization and the carrying capacity of the environment, making the development of urban agglomerations unsustainable. Therefore, in-depth exploration of the relationship between urbanization and the environment in urban agglomerations, promoting the coordinated development of urbanization and the environment, has become a hot topic in current relevant research areas and is also an important issue that urgently needs to be addressed in the process of urbanization in China.

Research on the relationship between urbanization and the eco-environment has mainly considered one-way influences, interactions, and coordinated development. Early research mainly focused on the one-way influence relationship between urbanization and the eco-environment, including the ecological changes caused by rapid urbanization and the constraints of the eco-environment on urbanization [12, 13]. As research progressed, attention was given to the interaction between the two and derived theoretical models of the interaction, such as the Environmental Kuznets Curve (EKC) and the biexponential curve [14, 15]. Quantitative empirical analysis of this interaction has been conducted using econometric models such as the system dynamics model, decoupling model, and GWR model [16, 17]. In recent years, the coupling relationship and coordinated sustainable development of urbanization and the eco-environment have become research hotspots and frontier fields. International organizations have maintained a high level of attention to this topic and have listed it as a key research topic [18–20]. The academic community has also conducted theoretical and practical explorations. The Fang Chuanglin team has proposed the coupling circle theory, remote coupling theory, and the concept and research framework of the coupling cube to analyze the coupling and coordinated relationship between urbanization and the eco-environment [21–23]. Other scholars have used relevant mathematical models to quantitatively evaluate the coordinated development of the two, among which the most widely applied is the Coupling Coordination Degree Model (CCD). Liao Chongbin [24], Liu Yaobin [25], and Qiao Biao [26] successively established a coupling coordination model between urbanization and the eco-environment based on the coefficient of variation, coupling coefficient, and trigonometric function principles. The model has been highly regarded by scholars and widely used for practical research. For example, Huang Jinchuan and others used CCD model to analyze the relationship between urbanization and the eco-environment in Kazakhstan and combined it with geographic detectors to study the main control factors affecting coordinated development [27]. Wang Zhenbo and others focused on the Beijing-Tianjin-Hebei urban agglomeration, applied CCD model to

analyze the relationship between urbanization and the eco-environment, and proposed a sustainable development path for green urbanization in the urban agglomeration [28]. Muhadaisi Ariken and others quantitatively evaluated the coordination relationship and spatiotemporal heterogeneity of urbanization and the eco-environment in provinces along the Silk Road in China through the integration of CCD and GTWR models [29].

The above research has played an important role in understanding the evolutionary relationship and coordinated development of urbanization and the eco-environment. But most of them focus on the study of the interaction between urbanization and eco-environment, and pay less attention to the positive interaction effect between the two. At the same time, there are more studies on the "one-to-one" interaction between urbanization and eco-environment at the system level, but there are few research results based on the "many-to-many" interaction between subsystems or elements. In addition, existing research lacks sufficient understanding of the interaction process and evolutionary mechanism of urbanization and eco-environment over time, and cannot explain the fundamental reasons for changes in the coordination relationship. The VAR model is an econometric model proposed by Christopher Sims in 1980, which is commonly used to analyze the dynamic interactive effects and dynamic shocks of variables in a time series with interdependencies and random disturbances. It has been widely applied in various fields [17, 30]. Compared with other econometric models, VAR models have the following main characteristics: first, there is a mutual influence relationship between multiple time series data; second, variables have intercorrelation or common driving factors; third, it models and predicts the dynamic relationship between multiple variables. As urbanization and eco-environment coupling system is a dynamic, complex, and nonlinear multiple feedback system, the elements in the system interact and constrain each other. Therefore, adopting VAR model to analyze the coupling relationship and interaction between urbanization and eco-environment can better reveal the dynamic evolution relationship and interaction mechanism between the two.

The Chengdu-Chongqing urban agglomeration is one of the five national-level urban agglomerations in China. It is an important demonstration area for the country's promotion of new urbanization and undertakes the national strategic tasks of building the "fourth pole" of China's future economic growth and the ecological security zone in the upper reaches of the Yangtze River. However, the Chengdu-Chongqing urban agglomeration faces severe ecological and environmental pressures while experiencing rapid economic and urbanization growth. Compared with other coastal urban agglomerations, such as Beijing-Tianjin-Hebei, the Yangtze River Delta, and the Greater Bay Area, the Chengdu-Chongqing urban agglomeration has more complex topography, climate conditions, and a fragile eco-environment, which frequently leads to natural disasters and serious soil erosion [31]. The industrial density in the Chengdu-Chongqing urban agglomeration is high, and the basin topography makes it difficult for industrial pollutants to dilute and diffuse, often resulting in severe regional air pollution [32]. Furthermore, the expansion of urban construction land and the protection of arable land have led to substantial conflicts; serious pollution in some tributary water environments and overall unsatisfactory environmental quality have developed [33]. In addition, due to the low efficiency of water, land, and energy resource utilization in traditional manufacturing industries, the Chengdu-Chongqing urban agglomeration faces increasingly severe resource constraints and pressures. There is an increasingly imbalanced situation between urbanization development and eco-environment carrying capacity. Urgently need to carry out research on the coupling relationship and interaction mechanism between urbanization and eco-environment, and then explore the path of coordinated sustainable development of the two.

Based on the above background, this article first analyzes the coupling relationship and interaction mechanism between urbanization and eco-environment from a theoretical

perspective, and then conducts empirical research on the Chengdu-Chongqing urban agglomeration in China as a typical region. The empirical analysis includes the following content: firstly, using ArcGIS and quadrant analysis method to analyze the spatial and temporal patterns and dynamic evolution characteristics of urbanization level and eco-environment quality in the Chengdu-Chongqing urban agglomeration; secondly, using the CCD model to explore the development rules and evolution characteristics of their coupling relationship; thirdly, using the VAR model to analyze the complex interaction between the influencing factors of the two systems, urbanization and eco-environment; finally, feasible policy recommendations are proposed based on the research results. The research results can provide scientific basis for the sustainable development of the Chengdu-Chongqing urban agglomeration, and also enrich the achievements in the research field of the coupling relationship between urbanization and eco-environment. This article may have some innovations in the following aspects: based on theoretical and empirical analysis from a dual perspective, it conducts an analysis of the "many-to-many" interaction of system components from the levels of interactive threats and benign interactions, revealing the interaction process and evolution mechanism between urbanization and eco-environment over time, and thereby explaining the fundamental reasons for changes in coupling relationships.

## 2 The coupling relationship and interaction mechanism between urbanization and eco-environment

Urbanization is one of the most important drivers of the Anthropocene and a crucial manifestation of human social development and evolution, encompassing a series of complex processes including large-scale population migration, urban land expansion, industrial structural adjustment, capital accumulation, cultural and consumption habit transformation [34, 35]. According to the manifestations and connotations of urbanization, the process of urbanization can be classified into four levels: population urbanization, spatial urbanization, economic urbanization, and social urbanization. Among them, population urbanization is the core of the entire system, economic urbanization is the driving force, spatial urbanization is the manifestation of changes in spatial patterns, and social urbanization reflects the diffusion of civilization and the level of people's living standards. The four subsystems of urbanization are interrelated and simultaneously have an impact on the eco-environment. The eco-environment is the natural foundation and support system necessary for human survival and reproduction, encompassing natural and artificial elements such as water, land, climate, energy, and greenery. It is the sum of various ecological factors and relationships that affect human survival and development [36]. Considering the relationship with urbanization, the eco-environment system can be divided into three subsystems: ecological conditions, ecological pressures, and ecological responses, known as the "Pressure-State-Response" framework (PSR). The eco-environment PSR framework reflects the interaction between human activities and the environment.

There is an objective dynamic coupling relationship between urbanization and eco-environment. This coupling relationship is essentially the sum of various nonlinear relationships formed by the interaction between urbanization elements such as population, economy, space, and society, and eco-environment elements such as ecological conditions, ecological pressures, and ecological responses. Interaction is the basis for the formation of coupling relationships. There is a strong interaction relationship between urbanization and eco-environment, forming a complex and nonlinear multiple feedback system, where the elements mutually promote and constrain each other (Fig 1).

The eco-environment provides natural resources and elements such as water, soil, air, energy, and minerals for urbanization, promoting the process of urbanization. Urbanization

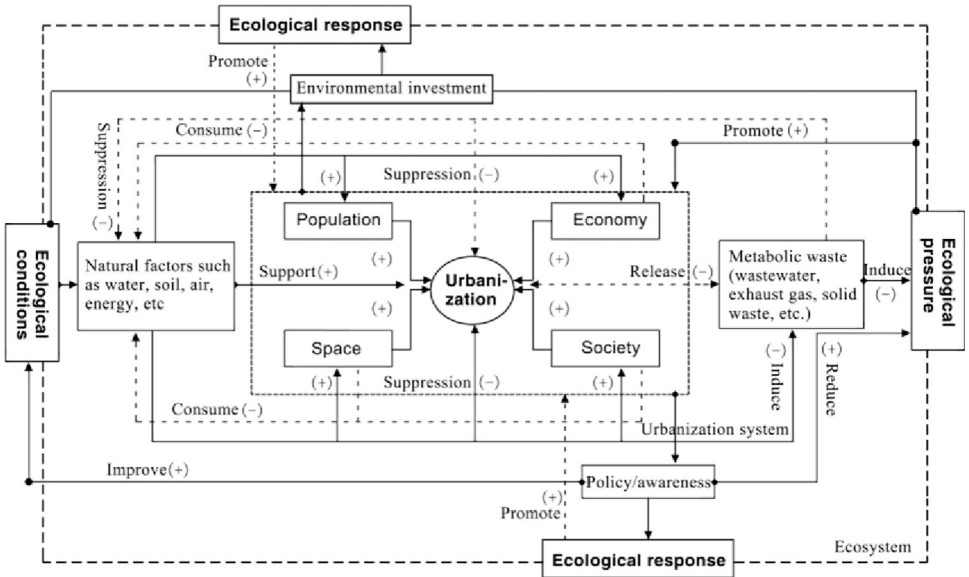

**Fig 1. The mechanism of interaction between urbanization and eco-environment.**

generates wastewater, exhaust gas, solid waste through processes such as population growth, land expansion, and energy consumption, exerting pressure on the eco-environment. This pressure increases the eco-environment's stress and reduces its carrying capacity, thereby inhibiting urbanization. At the same time, with the development of urbanization, economic growth, technological progress, improvement in urban management level, and enhancement of population quality and environmental awareness, measures such as policy intervention and increased investment in environmental protection funds are taken to improve the quality of the eco-environment. Therefore, the interaction between urbanization and the eco-environment, through mutual constraint and mutual promotion, continuously evolves in a coordinated development direction, ultimately promoting the complex system of urbanization and the eco-environment from a lower to a level of form evolution.

# 3 Materials and methods

## 3.1 Study area and data sources

**3.1.1 Study area.** The Chengdu-Chongqing Urban agglomeration (27°39′–109°03′N, 101°56′–109°15′E) is located in the southwestern part of China in the upstream region of the Yangtze River. It is one of five major national-level Urban agglomerations in China and is centered around Chongqing and Chengdu. The region has diverse landforms, including plains, hills, and mountains, and is characterized as a typical basin landform. The climate is subtropical humid monsoon. The administrative scope of the Chengdu-Chongqing Urban agglomeration includes 15 prefecture-level administrative units in Sichuan Province and 29 district/county-level administrative units in Chongqing, covering a total area of 185,000 km$^2$. In 2018, the permanent population was 95 million, accounting for 6.8% of the national total, and the regional GDP was 5.7 trillion yuan, accounting for 6.4% of the national total. The urbanization rate of the permanent population was 54.81%. For research on spatial integrity and data availability, nine central urban areas in Chongqing, including Yuzhong District, were merged into one basic unit. Additionally, areas such as Mianyang, Dazhou, and Ya'an in Sichuan Province and Kaizhou and Yunyang in Chongqing, which are not within the extent of the Chengdu-

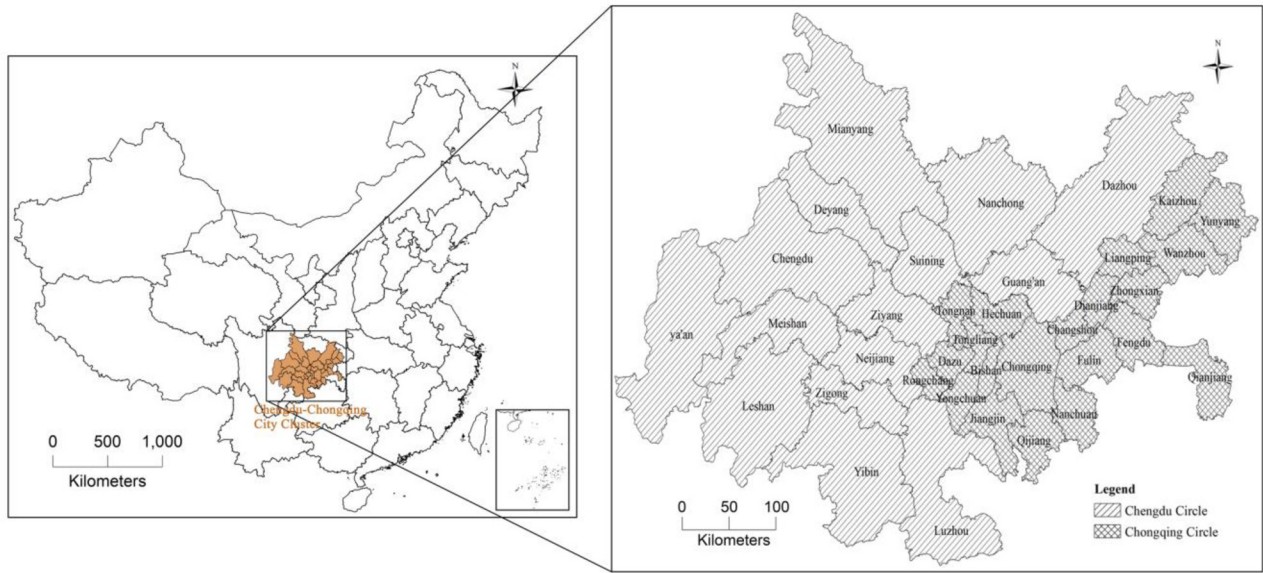

**Fig 2. Spatial distribution map of study area.** Reprinted from [(http://www.resdc.cn/DOI),2023.DOI:10.12078/2023010103] under a CC BY license, with permission from [Xu Xinliang], original copyright [January 2023]. Reprinted from [(http://www.resdc.cn/DOI),2023.DOI:10.12078/2023010101] under a CC BY license, with permission from [Xu Xinliang], original copyright [January 2023].

Chongqing Urban agglomeration, were also included in this study. Overall, 36 research units were identified (Fig 2).

**3.1.2 Data sources and preprocessing.** The data used in this study included geospatial and statistical data related to urbanization and the environment. The geospatial data was obtained from the Administrative Boundary Data of Chinese Counties provided by the Resource and Environment Science Data Center of the Chinese Academy of Sciences (https://www.resdc.cn/Datalist1.aspx?FieldTyepID=5,2 accessed on March 5, 2023). Administrative boundary data containing 36 constituent units were created via vector editing in ArcGIS 10.1 software. Statistical data on urbanization and environment indicators were obtained from various publications, including the China Urban Statistical Yearbook, China Urban and Rural Construction Statistical Yearbook, China County Statistical Yearbook, Sichuan Statistical Yearbook, Chongqing Statistical Yearbook, Sichuan environment Bulletin, Chongqing environment Bulletin, as well as city statistical yearbooks and national economic and social development statistical bulletins. The data cover the period from 2001 to 2020, and missing data were supplemented and improved using interpolation and trend analyses. The statistical data were classified and organized using Excel software to create the Chengdu-Chongqing Urban agglomeration Urbanization and Eco-environment Database.

## 3.2 Research methods

**3.2.1 Systematic index evaluation model.** *3.2.1.1 Construction of indicator system.* To obtain a comprehensive overview of the process of urbanization development, an evaluation index system for the urbanization development level was constructed based on four perspectives: population, economy, sociology, and space [37]. Population urbanization represents the foundation, economic urbanization represents the core components, social urbanization reflects the dissemination of civilization and the level of people's living standards, and regional urbanization reflects changes in land use structure and the establishment of transportation infrastructure; these offer a comprehensive picture of the level of urbanization development.

**Table 1. Comprehensive evaluation index system for urbanization and eco-environment based on population, economy, sociology, and space (PESS) and the Pressure-State-Response (PSR) model.**

| System | Subsystem | Evaluation indicators | Unit | Attribute |
|---|---|---|---|---|
| Urbanization system (U) | Population urbanization (PU) | Permanent population urbanization rate ($PU_1$) | % | + |
| | | Proportion of employees in the secondary and tertiary industries ($PU_2$) | % | + |
| | Regional urbanization (RU) | Urban quantity density ($RU_1$) | individual/hundred $km^2$ | + |
| | | Urban area density ($RU_2$) | % | + |
| | | Transportation network density ($RU_3$) | km/Ten thousand $km^2$ | + |
| | Economic urbanization (EU) | Per capita GDP ($EU_1$) | yuan | + |
| | | The proportion of value added from the secondary and tertiary industries to GDP ($EU_2$) | % | + |
| | | Per capita social fixed asset investment ($EU_3$) | yuan | + |
| | | Per capita industrial output value ($EU_4$) | yuan | + |
| | Social urbanization (SU) | Per capita total retail sales of consumer goods ($SU_1$) | yuan | + |
| | | Per capita disposable income of urban residents ($SU_2$) | yuan | + |
| | | Engel coefficient of urban residents ($SU_3$) | % | - |
| | | Number of cultural and artistic professionals per ten thousand people ($SU_4$) | people/ Ten thousand people | + |
| | | Number of medical and technical personnel per ten thousand people ($SU_5$) | people/ Ten thousand people | + |
| | | Number of hospital beds per ten thousand people ($SU_6$) | sheet/Ten thousand people | + |
| eco-environment system(E) | Ecological pressure (EP) | Average industrial wastewater discharge per unit of land area ($EP_1$) | Ton/$km^2$ | - |
| | | Average industrial gas emissions per unit of land area ($EP_2$) | Ten thousand cubic meters/$km^2$ | - |
| | | Average industrial solid waste generation per unit of land area ($EP_3$) | Ton/$km^2$ | - |
| | | Energy consumption per unit of GDP($EP_4$) | Ton of standard coal/Ten thousand yuan | - |
| | Ecological status (ES) | Per capita water resources ($ES_1$) | $m^3$ | + |
| | | Per capita arable land area ($ES_2$) | $hm^2$ | + |
| | | Forest coverage rate ($ES_3$) | % | + |
| | | Green coverage rate in built-up areas ($ES_4$) | % | + |
| | | Per capita park green space area in urban areas ($ES_5$) | $m^2$ | + |
| | Ecological response (ER) | Urban domestic wastewater treatment rate ($ER_1$) | % | + |
| | | Urban domestic waste harmless treatment rate ($ER_2$) | % | + |
| | | Industrial wastewater treatment compliance rate ($ER_3$) | % | + |
| | | Industrial exhaust gas removal rate ($ER_4$) | % | + |
| | | Comprehensive utilization rate of industrial solid waste ($ER_5$) | % | + |
| | | Proportion of environmental governance investment to GDP ($ER_6$) | % | + |

The Pressure-State-Response model was used to construct an evaluation index system for eco-environment quality, which represents the pressure faced by, current characteristics of, and response measures for the eco-environment [38]. A comprehensive evaluation index system for urbanization and the eco-environment is presented in Table 1.

*3.2.1.2 Data standardization and weighting.* To eliminate the influence of differences in the magnitude and dimensionality of each indicator on the calculation results, the indicators were standardized to reduce random interference. Different standardization formulas were used based on the attribute characteristics of the indicators.

$$A_{ij} = \frac{X_{ij}-\min(X_{ij})}{\max(X_{ij})-\min(X_{ij})} \left(" + "\text{indicator}\right), \ A_{ij} \frac{\max(X_{ij})-X_{ij}}{\max(X_{ij})-\min(X_{ij})} \left(" - "\text{indicator}\right) \quad (1)$$

where $i$ is the index number, $j$ is the year, $X_{ij}$ is the actual calculated value, and $\max(X_{ij})$ and $\min(X_{ij})$ are the maximum and minimum values of the $i$-th indicator, respectively. After standardization, larger values indicated better performance for all indicators.

A comprehensive weighting method, combining subjective and objective approaches, was used to assign weights to each indicator [39]. An analytic hierarchy process (AHP) was first used for subjective weighting, after which the entropy method was used for objective weighting. Finally, the weights from the subjective and objective approaches were integrated using the principle of minimum information entropy. The formula is expressed as follows:

$$w_i = \frac{\sqrt{w_{1i} \times w_{2i}}}{\sum_{i=1}^{n} \sqrt{w_{1i} \times w_{2i}}} \quad (2)$$

where $W_{1i}$ and $W_{2i}$ represent the weights calculated using AHP and entropy methods, respectively. $W_i$ represents the comprehensive weight of urbanization and eco-environment indicators. The weights of each evaluation indicator within the subsystem were first calculated according to the steps of the system index evaluation model, after which the weights of the subsystems were calculated. Table 2 presents the results of the study.

*3.2.1.3 Calculation of composite indices.* The composite index for both the urbanization and eco-environment subsystems was calculated using the linear weighting method. The formula is expressed as follows:

$$u = \sum_{i=1}^{n} w_i \times x_i, e = \sum_{j=1}^{m} w_j \times y_j \quad (3)$$

$$U = \sum_{i=1}^{n} W_i \times u, E = \sum_{j=1}^{m} W_j \times e \quad (4)$$

where $u$ and $e$ represent the evaluation values of the urbanization and eco-environment subsystems, respectively; $U$ and $E$ represent the comprehensive evaluation values of the urbanization and eco-environment systems, respectively; $x_i$ and $y_j$ represent the standardized values of the indicators; $w_i$ and $w_j$ represent the comprehensive weights of the indicators; $W_i$ and $W_j$ represent the comprehensive weights of the subsystems.

**3.2.2 Coupling coordination model.** As a physical concept, coupling refers to the phenomenon wherein two or more systems mutually influence each other owing to various interactions with the external environment. Because of the similarities in coupling relationships between systems, coupling coordination can be used to measure the interaction between urbanization and the eco-environment according to the following formula [40]:

$$C = \left\{ U \cdot E \quad / [( \quad U + E)/2]^2 \right\}^2 \quad (5)$$

$$T = \alpha \times U + \beta \times E \quad (6)$$

$$D = \sqrt{C \times T} \quad (7)$$

Where, $C$ is the degree of coupling, $D$ is the degree of coordinated development. The value of $C$ and $D$ ranges from 0 to 1, and a higher value indicates better coupling relationships and coordinated development between systems. $U$ and $E$ represent the urbanization and eco-environment comprehensive evaluation indexes, while $T$ stands for the comprehensive development index of urbanization and eco-environment system. $\alpha$ and $\beta$ are undetermined weights

**Table 2. Subsystem and indicator weights for urbanization and environment.**

| System | Subsystem | Subsystem weight | | | Evaluation indicators | Indicator weight | | |
|---|---|---|---|---|---|---|---|---|
| | | Analytic hierarchy process (AHP) method weights | Entropy method weights | Composite weights | | AHP method weights | Entropy method weights | Composite weights |
| Urbanization system(U) | Population urbanization (PU) | 0.25 | 0.2719 | 0.2614 | $PU_1$ | 0.6667 | 0.6258 | 0.6466 |
| | | | | | $PU_2$ | 0.333 | 0.3742 | 0.3534 |
| | Regional urbanization (RU) | 0.25 | 0.2405 | 0.2458 | $RU_1$ | 0.1818 | 0.353 | 0.2584 |
| | | | | | $RU_2$ | 0.5455 | 0.4503 | 0.5054 |
| | | | | | $RU_3$ | 0.2727 | 0.1967 | 0.2362 |
| | Economic urbanization (EU) | 0.25 | 0.29 | 0.2699 | $EU_1$ | 0.4029 | 0.2331 | 0.3192 |
| | | | | | $EU_2$ | 0.2728 | 0.1703 | 0.2245 |
| | | | | | $EU_3$ | 0.1291 | 0.2908 | 0.2018 |
| | | | | | $EU_4$ | 0.1952 | 0.3059 | 0.2545 |
| | Social urbanization (SU) | 0.25 | 0.1977 | 0.2229 | $SU_1$ | 0.0812 | 0.2833 | 0.1652 |
| | | | | | $SU_2$ | 0.3129 | 0.0976 | 0.1904 |
| | | | | | $SU_3$ | 0.0899 | 0.0575 | 0.0783 |
| | | | | | $SU_4$ | 0.1286 | 0.2813 | 0.2072 |
| | | | | | $SU_5$ | 0.2594 | 0.1861 | 0.2393 |
| | | | | | $SU_6$ | 0.1279 | 0.0942 | 0.1196 |
| eco-environment system(E) | Ecological pressure (EP) | 0.3532 | 0.2799 | 0.317 | $EP_1$ | 0.1993 | 0.1972 | 0.2004 |
| | | | | | $EP_2$ | 0.3244 | 0.2596 | 0.2934 |
| | | | | | $EP_3$ | 0.1088 | 0.2122 | 0.1536 |
| | | | | | $EP_4$ | 0.3675 | 0.331 | 0.3526 |
| | Ecological status (ES) | 0.4475 | 0.5739 | 0.5109 | $ES_1$ | 0.2687 | 0.6608 | 0.4607 |
| | | | | | $ES_2$ | 0.0874 | 0.0629 | 0.0811 |
| | | | | | $ES_3$ | 0.3373 | 0.1148 | 0.2152 |
| | | | | | $ES_4$ | 0.1699 | 0.0821 | 0.1291 |
| | | | | | $US_5$ | 0.1367 | 0.0794 | 0.1139 |
| | Ecological response (ER) | 0.1993 | 0.1462 | 0.1721 | $UR_1$ | 0.1485 | 0.2464 | 0.1943 |
| | | | | | $UR_2$ | 0.1166 | 0.1262 | 0.1232 |
| | | | | | $UR_3$ | 0.1729 | 0.0844 | 0.1227 |
| | | | | | $UR_4$ | 0.2151 | 0.176 | 0.1976 |
| | | | | | $UR_5$ | 0.0791 | 0.0912 | 0.0863 |
| | | | | | $UR_6$ | 0.2678 | 0.2758 | 0.276 |

representing the contribution shares of urbanization and eco-environment, respectively. In this study, $\alpha = \beta = 0.5$ is considered.

To further analyze the relative relationship between the development of urbanization and eco-environment systems, the relative development degree model was introduced, and is calculated as follows:

$$\varepsilon = E/U \tag{8}$$

where $\varepsilon$ represents the relative degree of development. When $\varepsilon \in (0, 0.9)$, it indicates that the level of urbanization development exceeds the carrying capacity of the eco-environment; when $\varepsilon \in (0.9, 1.1)$, it indicates that the level of urbanization development is equivalent to the carrying capacity of the eco-environment, implying basic synchronous development; when $\varepsilon \in (1.1, \infty)$, it indicates that the level of urbanization development lags behind the carrying capacity of the eco-environment.

**Table 3. Coupling relationship and coordinated development types of urbanization and environment.**

| Coupling relationship | D value | ε value | System status | Characteristics of coordinated and integrated development |
|---|---|---|---|---|
| low-level coupling | $0 \leq D < 0.3$ | —— | disordered state(I) | The level of urbanization is low, and there is relatively little attention paid to the eco-environment. (I) |
| antagonistic stage | $0.3 \leq D < 0.6$ | $0 < ε \leq 0.9$ | low steady state(II) | The pace of urbanization development is fast, exceeding the carrying capacity of the eco-environment. (II-1) |
|  |  | $0.9 < ε < 1.1$ |  | The level of urbanization development is commensurate with the carrying capacity of the environment.(II-2) |
|  |  | $1.1 \leq ε < \infty$ |  | The pace of urbanization development is slow, and the environment can sustain in the short term. (II-3) |
| adjustment phase | $0.6 \leq D < 0.8$ | $0 < ε \leq 0.9$ | medium steady state(III) | The pace of urbanization development is fast, and the system boundaries align rapidly.(III-1) |
|  |  | $0.9 < ε < 1.1$ |  | The pace of urbanization development is moderate, and the system boundaries are dynamically aligned.(III-2) |
|  |  | $1.1 \leq ε < \infty$ |  | The pace of urbanization development is slow, and the system boundaries are slow to align.(III-3) |
| high-level coupling | $0.8 \leq D \leq 1$ | $0 < ε \leq 0.9$ | high steady state (IV) | Urbanization develops excessively, causing impacts on the environment.(IV-1) |
|  |  | $0.9 < ε < 1.1$ |  | Urbanization and environment develop in a coordinated manner, with significant constraints from the carrying capacity of the environment.(IV-2) |
|  |  | $1.1 \leq ε < \infty$ |  | The pace of urbanization development slows down, and the environment is generally sustainable. (IV-3) |

Based on the theory of multiple stable states in ecological systems [41] and combining the coupling coordination and relative development degrees, the coupling relationship and development types of urbanization and the eco-environment were classified into ten basic types in four stages: disordered, low stable, medium stable, and highly stable (Table 3).

**3.2.3 VAR model.** The VAR model is a time-series analysis method used to study the predictions of related time-series systems and the dynamic impact of random disturbances on variable systems. It evaluates the dynamic relationships among all endogenous variables in a model by regressing their lagged values. Urbanization and eco-environment systems are complex and nonlinear multi-feedback systems in which elements interact with and constrain each other. The VAR model is well-suited for analyzing such complex relationships, and the analysis process is as follows.

*3.2.3.1 Model Construction.* This study examined the impact of urbanization on the eco-environment from four perspectives: population, space, economy, and society. The feedback of environmental changes on the urbanization process was analyzed with regard to three aspects: ecological pressure, ecological status, and ecological response. Therefore, a VAR model was established for the multivariate system of urbanization (PU, RU, EU, and SU) and the eco-environment (EP, ES, and ER):

$$Y_{it} = \sum_{j=1}^{p} \beta_j Y_{it-j} + \gamma_t + u_t \tag{9}$$

where $Y_{it}$ represents the endogenous variable, $i$ represents the Urbanization and eco-environment indicators, $p$ represents the lag order, $\beta_i$ represents the coefficient matrix, $\gamma_t$ represents the time–effect vector, and $u_t$ represents the disturbance term.

*3.2.3.2 Model validity verification.* The VAR model effectiveness test mainly examines the stationarity and stability of the system variables. This article uses unit root, cointegration relationship, and AR characteristic roots to test the effectiveness of the model, while also using the Granger causality analysis to examine the mutual influence relationship between urbanization and eco-environmental variables.

*3.2.3.3 Impulse response analysis.* The impulse response function shows how a shock in one endogenous variable is transmitted through the dynamic structure of the VAR model to all other endogenous variables and eventually feeds back to itself. Its function is expressed as follows:

$$Y_{t+s} = U_{t+s} + \Psi_1 U_{t+s-1} + \Psi_2 U_{t+s-2} + \ldots + \Psi_s U_t (t = 1, 2, \ldots, T) \tag{10}$$

$$\Psi_s = \vartheta Y_{t+s} / \vartheta U_t \tag{11}$$

where $Y_{t+s}$ represents the endogenous vector, $U_t$ represents the random error term, and $\Psi_s$ represents the impulse response vector.

*3.2.3.4 Variance decomposition.* The forecast errors of the endogenous variables in the model were decomposed based on their causes. The impact of each information shock on the endogenous variables in the model was measured by calculating the variance values, and the relative contribution ratio of each variable was then calculated. The forecast errors for the first s periods in the VAR($p$) model are as follows:

$$\varepsilon_{t+s} + \lambda_1 \varepsilon_{t+s-1} + \lambda_2 \varepsilon_{t+s-2} + \ldots + \lambda_2 \varepsilon_{t+s-2} + \ldots + \lambda_{s-1} \varepsilon_{t-1} \tag{12}$$

Where, $\lambda_{s-1}$ represents the lagged reflection of the s-1 period.

## 4 Results

### 4.1 Spatial and temporal evolution characteristics of urbanization and eco-environment

Based on the results of the comprehensive index calculations, the urbanization and eco-environment index of the Chengdu-Chongqing urban agglomeration were divided into five levels using the natural breaks classification method in ArcGIS (jenks): low-value, low-medium value, medium-value, medium-high value, and high-value areas [42] (Table 4). Four time points (2001, 2008, 2014, and 2020) were selected to analyze the spatial and temporal evolution patterns of urbanization and eco-environment indices, as well as the dynamic relationship between the two.

**4.1.1 Evolution characteristics of urbanization index.** Overall, the urbanization index of the Chengdu-Chongqing urban agglomeration is relatively low, but the development speed is fast and showing an upward trend year by year. There are significant spatial distribution differences in the urbanization index among the 36 cities, forming a high-value spatial agglomeration area centered around Chengdu and Chongqing, and a low-value agglomeration area mainly composed of cities in the northeastern part of the urban agglomeration, such as Dazhou, Kaizhou, and Yunyang (Fig 3).

**Table 4. Chengdu-Chongqing urban agglomeration urbanization and eco-environment index classification table.**

| level | Type | Index range | |
|---|---|---|---|
| | | Urbanization index | eco-environment Index |
| I | low-value area | 0.0346–0.1619 | 0.2406–0.3842 |
| II | low-medium value area | 0.1620–0.2668 | 0.3843–0.4685 |
| III | medium-value area | 0.2669–0.3857 | 0.4686–0.5401 |
| IV | medium-high value area | 0.3858–0.5468 | 0.5402–0.6058 |
| V | high-value area | 0.5469–0.8539 | 0.6059–0.7434 |

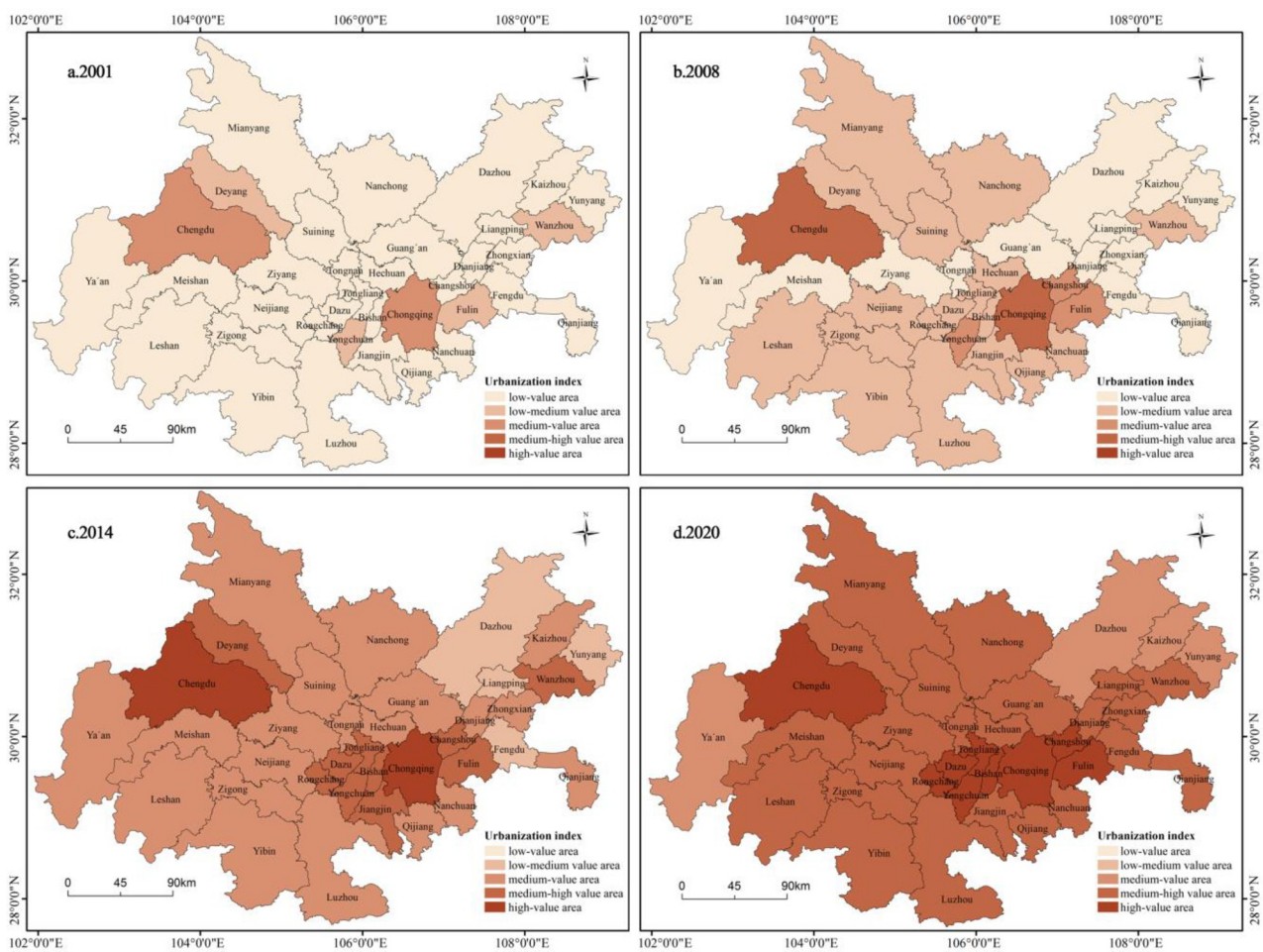

**Fig 3. Spatial and temporal evolution characteristics of urbanization index.** Reprinted from [(http://www.resdc.cn/DOI),2023.DOI:10.12078/2023010101] under a CC BY license, with permission from [Xu Xinliang], original copyright [January 2023].

Chengdu and Chongqing, as the two major central cities of the Chengdu-Chongqing urban agglomeration, have a good foundation for urbanization. They leverage the locational advantages of central cities to generate a strong "siphon effect," attracting a large population, high-quality market resources, technological talent, and financial capital. The urban areas continue to expand outward, with continuous improvements in urban infrastructure and social public services, leading to a consistently high urbanization index. At the same time, they also have a "radiation effect" on surrounding cities, promoting the continuous improvement of their urbanization levels, collectively forming high-value clusters of urbanization indexes.

The urbanization index of cities such as Dazhou, Kaizhou, and Yunyang has always been low, with small increases. The main reason for this is that these cities are located on the northeast edge of urban agglomerations, far from the two major central cities of Chengdu and Chongqing, and most of them are mountainous areas with weak urbanization foundations. These areas have low levels of industrial development, with the industrial sector still in its early stages. Although the tourism industry has some development, the backward infrastructure construction and limited public service capabilities prevent the tourism industry from becoming the main support for urbanization development. In addition, urban economic development mainly relies on national and local investments, with low total retail sales of social

consumer goods and limited ability to drive economic growth through consumption, leading to slow urban construction.

**4.1.2 Evolution characteristics of eco-environment index.** Compared to the urbanization index, the eco-environment of the Chengdu-Chongqing urban agglomeration is relatively better, but the absolute level is still low. Over the past twenty years, the eco-environment index of the 36 cities has shown a gradual increase, with a relatively fast growth rate. In terms of spatial distribution characteristics, there are significant spatial differences in the changes of the eco-environment index among the cities, roughly characterized by high values around the periphery and low values in the central region (Fig 4).

Considering the practical development of the Chengdu-Chongqing urban agglomeration, it is found that there is a significant negative correlation between the eco-environment and economic development (urbanization). Specifically, economically underdeveloped areas tend to have better ecological quality, while areas with higher levels of economic development tend to have poorer ecological quality. Furthermore, this negative correlation becomes more significant over time. For example, cities such as Ya'an, Yunyang, and Qianjiang have relatively lagging economic development, resulting in less impact on the eco-environment and better ecological quality. On the other hand, cities like Chengdu, Chongqing, and Deyang face

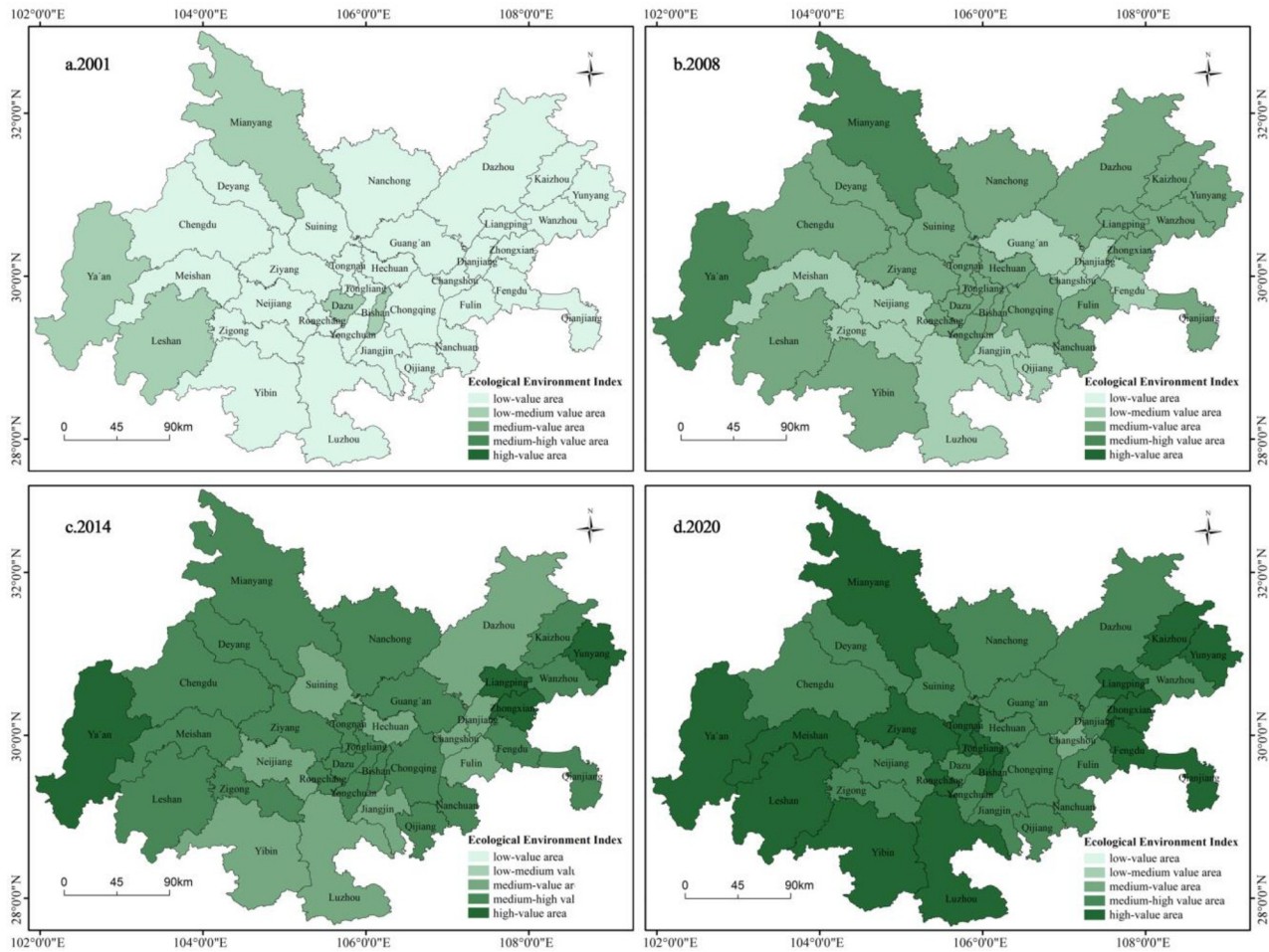

**Fig 4. Spatial and temporal evolution characteristics of eco-environment index.** Reprinted from [(http://www.resdc.cn/DOI),2023.DOI:10.12078/ 2023010101] under a CC BY license, with permission from [Xu Xinliang], original copyright [January 2023].

greater pressure on the eco-environment due to economic development. Although a significant amount of funds have been invested in environmental pollution control and eco-facility construction during the development process, the damage caused by economic development to the eco-environment surpasses its recovery speed, resulting in a poor ecological quality. Additionally, there are also cases where certain cities have relatively lagging economic development but poor ecological quality, such as Nanchong, Suining, and Dianjiang. Despite the lesser pressure on the eco-environment from economic and social development in these areas, they suffer from extensive development and insufficient investment in environmental protection. Moreover, the weak awareness of eco-environment protection among residents contributes to the poor ecological quality.

**4.1.3 Dynamic relationship between urbanization and eco-environment evolution.** Using urbanization and environment indices as the x- and y-axes, respectively, and mapping the cities onto the same quadrant and comparing them with the 45˚ contour line, the dynamic relationship between urbanization and environment evolution was analyzed for 2001, 2008, 2014, and 2020 (Fig 5). From 2001–2020, all cities gradually moved from the lower-left quadrant towards the vicinity of the contour line in the upper-right quadrant. Additionally, some cities crossed the contour line, indicating that both urbanization and environmental quality in the Chengdu-Chongqing urban agglomeration gradually increased. However, the growth rate of the environment index was lower than that of the urbanization index, suggesting increasing pressure on the environment. In 2001, the urbanization and environment indices of Chongqing were approximately 0.3501 and 0.3490, respectively, on the contour line. The remaining 35 cities were all located to the left of the contour line, with most of them close to the y-axis, indicating that the environment index was higher than the urbanization index. By 2008, both the urbanization and environment indices had gradually increased. Chongqing's environmental index grew faster than its urbanization index. Although all 36 cities were still located to the left of the contour line, their overall positions had shifted towards the right, indicating that during this period, the urbanization index grew faster than the environment index. In 2014, 34 cities moved closer to the contour line, while Chengdu and Chongqing had already crossed to the right of the contour line, indicating that the urbanization index had surpassed the environment index, indicating notable environment pressure. By 2020, the cities of Bishan, Changshou, Fuling, Yongchuan, and Dazu also crossed to the right of the contour line. Chengdu and Chongqing moved further away from the contour line, while other cities approached it, indicating a narrowing gap between the urbanization and environment indices in the Chengdu-Chongqing urban agglomeration.

## 4.2 Urbanization and eco-environment coupling relationship and evolution types

**4.2.1 Development patterns of coupling relationship.** Fig 6 shows the evolving characteristics of the coupling relationship and development status between urbanization and the environment in the Chengdu-Chongqing urban agglomeration from 2001 to 2020. Overall, the degree of coupling between urbanization and the eco-environment system is higher than the index of coordinated development, but the evolutionary trends of both exhibit a similar pattern of initially decreasing and then gradually increasing. In terms of coupling relationship, the overall degree of coupling between urbanization and the eco-environment system is high, and over the past twenty years, the coupling relationship has undergone a process of antagonism, coordination, and high-level coupling. In terms of the development index of the system's coordinated development, the urbanization and eco-environment system had a relatively low development index and underwent a process of transitioning from a low steady state to a

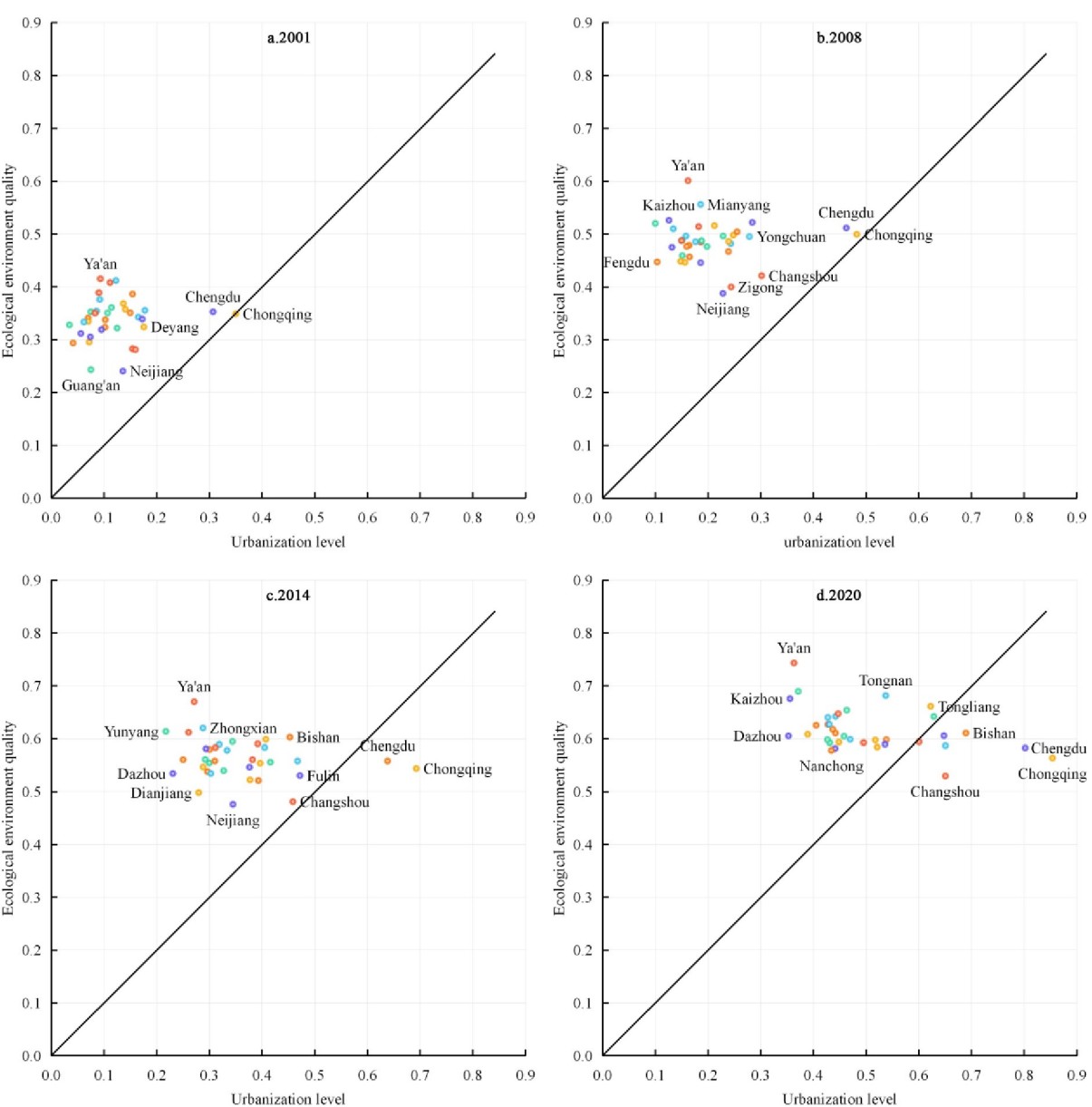

**Fig 5. Dynamic relationship between urbanization and eco-environment evolution.**

medium steady state over the past twenty years: from 2001 to 2015, the system was in a low steady state stage, and the development index exhibited a "V" shaped pattern; from 2015 to 2020, the development index of the urbanization and eco-environment system continued to increase, and the system state transitioned to a medium steady state.

**4.2.2 Dynamic evolution characteristics of coupling types.** From 2001 to 2020, the coupled development between urbanization and the eco-environment in the Chengdu-Chongqing urban agglomeration showed a positive trend, gradually transitioning from a state of coexistence between disorder and low stability to a state of coexistence between low and medium stability, and further to a state of coexistence between low, medium, and high stability. The main coupling types were low and medium stability, with some cities exhibiting disorder and high stability over a few years (Fig 7).

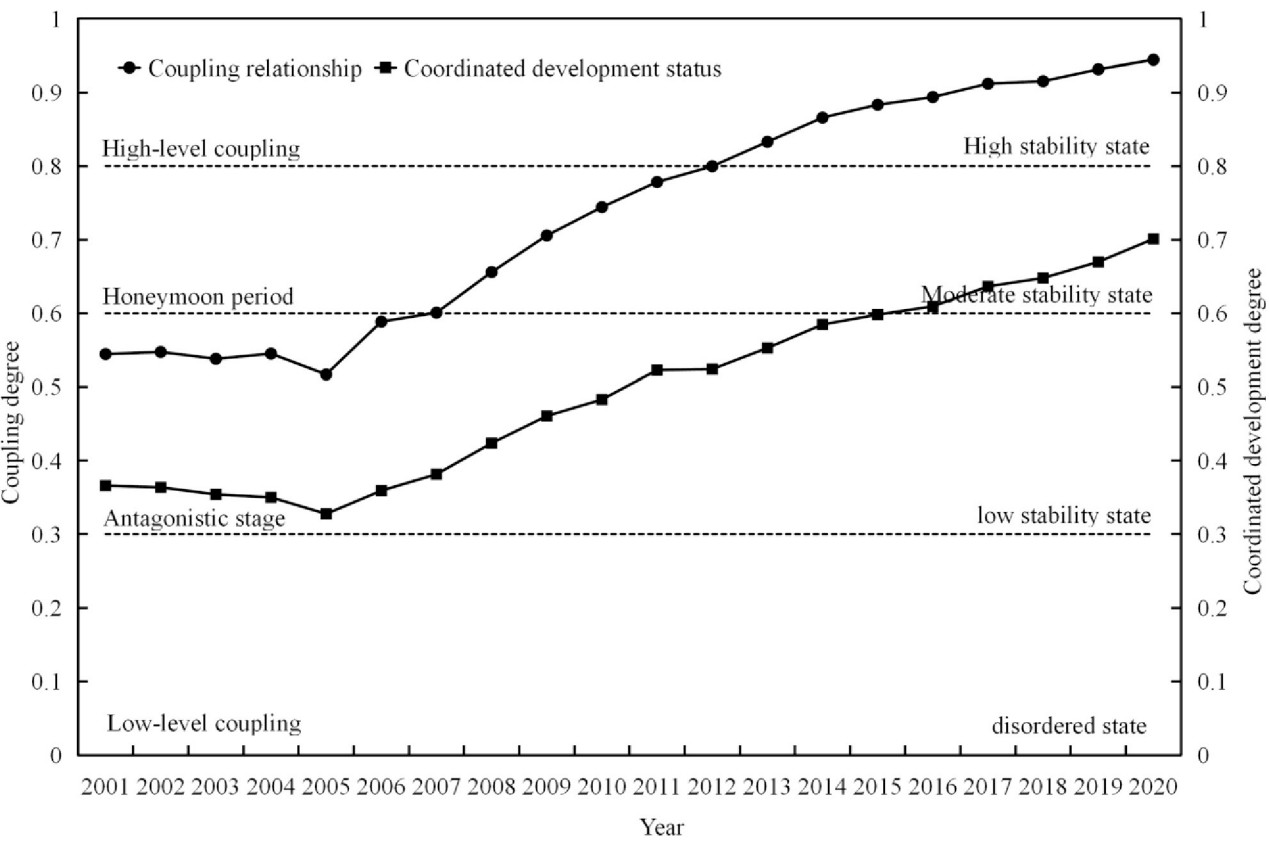

**Fig 6. Coupling relationship and coordinated development types between urbanization and eco-environment in the Chengdu-Chongqing urban agglomeration.**

Coexistence between disorder and low stability (2001–2007): During this period, the level of urbanization was low, the development was slow, and the attention to the eco-environment was minimal. Meishan, Guang'an, Qianjiang, and Kaizhou gradually transitioned from an unorganized state to a low steady state, while Fengdu and Yunyang cities remained in an unorganized development stage.

Coexistence between low and medium stability (2008–2015): During this period, the development momentum of Chengdu and Chongqing far exceeded that of other cities, consistently maintaining a stable stage. Meanwhile, eight cities including Meishan, Dazhou, and Ya'an remained in a low stable development stage. The urbanization level of Zigong, Deyang, and Wanzhou increased rapidly, gradually transitioning towards a stable stage. During this period, the speed of urbanization development accelerated, while the relative development index noticeably declined. After a brief period of synchronized development, Chengdu and Chongqing began to show a more obvious lag in their eco-environment. On the other hand, Fuling, Changsh, and Yongchuan gradually transitioned from a laggingization type to a synchronized development type.

Coexistence between low, medium, and high stability (2016–2020): During this period, Chongqing, Chengdu, Bishan, and Tongliang have successively entered a phase of high and stable development; cities such as Dazhou, Ya'an, and Kaizhou have transitioned from a phase of low stability to a phase of medium stability. At the same time, urbanization has rapidly developed, with cities such as Fuling, Yongchuan, Dazu, and Rongchang gradually catching up with the level of eco-environment, showing a synchronized development status; whereas cities

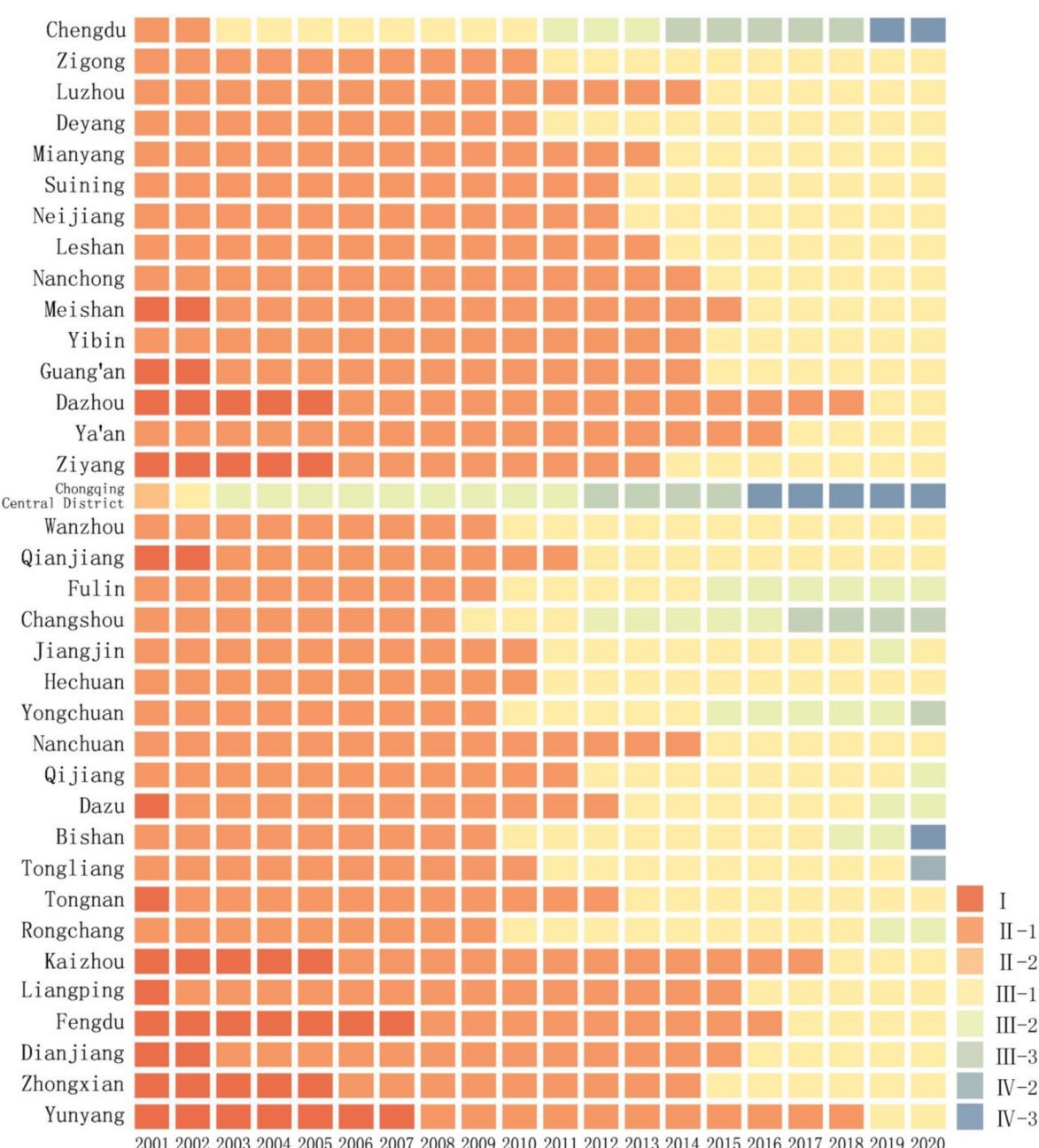

**Fig 7. Dynamic evolution of coupling and coordinated development between urbanization and eco-environment in the Chengdu-Chongqing urban agglomeration from 2001 to 2020.**

like Chengdu, Chongqing, and Changshou have exceeded the carrying capacity of the eco-environment, leading to increased environmental pressure.

## 4.3 Interaction between urbanization and eco-environment

**4.3.1 Model validation results.** First, the original series was transformed using a natural logarithm, and the ADF method was used to test the stationarity of the model variables. The test results showed that lnPU, lnEU, lnSU, and lnER are stationary series, whereas lnRU, lnES,

**Table 5. Unit root test results.**

| Variable | Differential order | ADF | p | Critical value | | | AIC | BIC | Conclusion |
|---|---|---|---|---|---|---|---|---|---|
| | | | | 1% | 5% | 10% | | | |
| lnPU | 0 | -3.000 | 0.035 | -4.138 | -3.155 | -2.714 | -89.434 | -85.070 | stability |
| lnRU | 0 | -0.699 | 0.847 | -4.223 | -3.189 | -2.730 | -81.269 | -77.290 | unstable |
| | 1 | -5.102 | 0.000 | -4.223 | -3.189 | -2.730 | -78.891 | -75.310 | stability |
| lnEU | 0 | -6.683 | 0.000 | -4.223 | -3.189 | -2.730 | -71.209 | -67.230 | stability |
| lnSU | 0 | -3.883 | 0.002 | -4.223 | -3.189 | -2.730 | -63.389 | -59.411 | stability |
| lnEC | 0 | -0.950 | 0.771 | -4.012 | -3.104 | -2.691 | -52.993 | -48.519 | unstable |
| | 1 | -1.028 | 0.743 | -4.012 | -3.104 | -2.691 | -53.296 | -49.462 | unstable |
| lnEP | 0 | -0.674 | 0.853 | -4.012 | -3.104 | -2.691 | -79.640 | -75.166 | unstable |
| | 1 | -3.584 | 0.006 | -3.924 | -3.068 | -2.674 | -94.897 | -91.806 | stability |
| lnER | 0 | -3.740 | 0.004 | -4.069 | -3.127 | -2.702 | -58.562 | -54.043 | stability |

and lnEP become stationary series after the first differencing. Subsequently, a cointegration test was conducted on the three non-stationary series variables, lnRU, lnES, and lnEP. Based on the LR statistics and information criteria such as AIC, BIC, FPE, and HQIC, the lag order of the model was determined to be 2. The Johansen cointegration test was used for cointegration analysis, and the trace statistic was used as the criterion for judging the results of the Johansen cointegration test. Third, use AR feature root to check the stability and effectiveness of VAR model coefficients. Finally, the Granger causality analysis is used to examine the interrelationships between urbanization and eco-environmental variables. The relevant test results are shown in Tables 5–7.

Unit root test results show that the four variables lnPU, lnEU, lnSU, and lnER are stationary sequences, while the three variables lnRU, lnES, and lnEP become stationary sequences after first differencing. Cointegration analysis results indicate a long-term stable relationship between urbanization and eco-environment, as confirmed by AR eigenvalue test. Granger

**Table 6. Cointegration and eigenvalue test results.**

| Equation | Lag order | Null Hypothesis | Eigenvalue | Trace | Critical value | | |
|---|---|---|---|---|---|---|---|
| | | | | | 10% | 5% | 1% |
| M1 | 2 | No cointegration relationship | 0.985 | 114.016 | 44.493 | 47.855 | 54.681 |
| | | At most 1 cointegration relationship | 0.753 | 42.993 | 27.067 | 29.796 | 35.463 |
| | | At most 2 cointegration relationships | 0.581 | 19.249 | 13.429 | 15.494 | 19.935 |
| | | At most 3 cointegration relationships | 0.231 | 4.457 | 2.705 | 3.841 | 6.635 |
| M2 | 2 | No cointegration relationship | 0.963 | 117.130 | 51.649 | 55.246 | 62.520 |
| | | At most 1 cointegration relationship | 0.920 | 61.126 | 32.065 | 35.012 | 41.081 |
| | | At most 2 cointegration relationships | 0.441 | 18.117 | 16.162 | 18.398 | 23.148 |
| | | At most 3 cointegration relationships | 0.383 | 8.219 | 2.705 | 3.841 | 6.635 |
| M3 | 2 | No cointegration relationship | 0.969 | 119.223 | 51.649 | 55.246 | 62.520 |
| | | At most 1 cointegration relationship | 0.879 | 60.410 | 32.065 | 35.012 | 41.081 |
| | | At most 2 cointegration relationships | 0.683 | 24.563 | 16.162 | 18.398 | 23.148 |
| | | At most 3 cointegration relationships | 0.256 | 5.031 | 2.705 | 3.841 | 6.635 |
| M4 | 2 | No cointegration relationship | 0.867 | 74.514 | 44.493 | 47.855 | 54.681 |
| | | At most 1 cointegration relationship | 0.752 | 40.247 | 27.067 | 29.796 | 35.463 |
| | | At most 2 cointegration relationships | 0.547 | 16.543 | 13.429 | 15.494 | 19.935 |
| | | At most 3 cointegration relationships | 0.165 | 3.070 | 2.705 | 3.841 | 6.635 |

**Table 7. Granger causality analysis results.**

| Null Hypothesis | F-value | p-value | df 1 | df 2 |
|---|---|---|---|---|
| lnPU is not the Granger reason for lnEP | 4.023 | 0.044** | 2 | 13 |
| lnPU is not the Granger reason for lnES | 12.196 | 0.001*** | 2 | 13 |
| lnPU is not the Granger reason for lnER | 4.085 | 0.043** | 2 | 13 |
| lnRU is not the Granger reason for lnEP | 4.56 | 0.038** | 2 | 13 |
| lnRU is not the Granger reason for lnES | 6.694 | 0.010** | 2 | 13 |
| lnRU is not the Granger reason for lnER | 5.363 | 0.020** | 2 | 13 |
| lnEU is not the Granger reason for lnEP | 2.519 | 0.093* | 2 | 13 |
| lnEU is not the Granger reason for lnES | 6.123 | 0.013** | 2 | 13 |
| lnEU is not the Granger reason for lnER | 4.906 | 0.026** | 2 | 13 |
| lnSU is not the Granger reason for lnEP | 3.285 | 0.075* | 2 | 13 |
| lnSU is not the Granger reason for lnES | 3.688 | 0.054* | 2 | 13 |
| lnSU is not the Granger reason for lnER | 7.763 | 0.006*** | 2 | 13 |
| lnEP is not the Granger reason for lnPU | 3.328 | 0.059* | 2 | 13 |
| lnEP is not the Granger reason for lnRU | 3.038 | 0.077* | 2 | 13 |
| lnEP is not the Granger reason for lnEU | 3.831 | 0.045** | 2 | 13 |
| lnEP is not the Granger reason for lnSU | 2.464 | 0.097* | 2 | 13 |
| lnES is not the Granger reason for lnPU | 0.256 | 0.086* | 2 | 13 |
| lnES is not the Granger reason for lnRU | 6.652 | 0.010** | 2 | 13 |
| lnES is not the Granger reason for lnEU | 3.24 | 0.072* | 2 | 13 |
| lnES is not the Granger reason for lnSU | 3.759 | 0.046** | 2 | 13 |
| lnER is not the Granger reason for lnPU | 7.134 | 0.009*** | 2 | 13 |
| lnER is not the Granger reason for lnRU | 5.155 | 0.032** | 2 | 13 |
| lnER is not the Granger reason for lnEU | 5.835 | 0.019** | 2 | 13 |
| lnER is not the Granger reason for lnSU | 2.466 | 0.087* | 2 | 13 |

**Note:** * $p < 0.1$
** $p < 0.05$
*** $p < 0.01$

causality analysis shows that there is a mutual influence relationship between urbanization and eco-environment variables. These test results suggest that the parameters of the VAR model are stable, allowing for impulse response and variance analysis.

**4.3.2 Impulse response analysis.** *4.3.2.1 Overall characteristics of impulse response.* Based on the four constructed relationship models, a pairwise analysis of the seven variables yielded 24 impulse response results, representing the interactions between the subsystems of urbanization and the eco-environment (Table 8). According to Table 8, the seven impulse response results are positive, indicating a beneficial promotion effect between the variables. Eight impulse response results were negative, indicating a restraining effect between variables. Additionally, the nine impulse response results exhibited both positive and negative changes, suggesting that the interaction between the variables can transition between promotion and inhibition over time. Furthermore, in terms of the cumulative impulse response, the maximum and minimum absolute values were observed in "lnEU→lnEP" and "lnEP→lnRU" respectively, indicating that ecological pressure has the most significant impulse response on economic urbanization, whereas regional urbanization has the weakest impulse response on ecological pressure.

*4.3.2.2 Analysis of impulse response process.* (1) Impulse response between population urbanization and the environment(Fig 8A and 8B). Urbanization of the population has a clear

**Table 8. Impulse response results between urbanization and eco-environment.**

| Model | Type | Variable | Direction | Trend | Impulse Response Cumulative |
|---|---|---|---|---|---|
| M1 | The impulse response of the environment to PU | lnPU→lnES | Negative | Decrease | -0.0067 |
| | | lnPU→lnEP | Positive | Increase-decrease | 0.0168 |
| | | lnPU→lnER | Negative | Decrease | -0.0101 |
| | The impulse response of PU to the environment | lnES→lnPU | Positive and negative fluctuations | Fluctuation-decrease | -0.0198 |
| | | lnEP→lnPU | Negative | Decrease | -0.0011 |
| | | lnER→lnPU | Positive | Decrease | 0.0447 |
| M2 | The impulse response of the environment to RU | lnRU→lnES | Positive-negative | Fluctuation-decrease | -0.0562 |
| | | lnRU→lnEP | Positive | Decrease | 0.0072 |
| | | lnRU→lnER | Negative | Fluctuation-decrease | -0.0487 |
| | The impulse response of RU to the environment | lnES→lnRU | Positive-negative | Fluctuation | -0.0072 |
| | | lnEP→lnRU | Positive and negative fluctuations | Fluctuation | 0.0010 |
| | | lnER→lnRU | Positive | Fluctuation-decrease | 0.0294 |
| M3 | The impulse response of the environment to EU | lnEU→lnES | Negative | Increase-decrease | -0.1690 |
| | | lnEU→lnEP | Positive | Increase-decrease | 0.2457 |
| | | lnEU→lnER | Positive and negative fluctuations | Fluctuation-decrease | 0.0297 |
| | The impulse response of EU to the environment | lnES→lnEU | Positive-negative | Fluctuation-decrease | -0.0472 |
| | | lnEP→lnEU | Negative | Decrease | -0.0029 |
| | | lnER→lnEU | Positive and negative fluctuations | Fluctuation-decrease | 0.0349 |
| M4 | The impulse response of the environment to SU | lnSU→lnES | Positive and negative fluctuations | Fluctuation-decrease | -0.0226 |
| | | lnSU→lnEP | Positive | Increase-decrease | 0.0565 |
| | | lnSU→lnER | Negative | Increase-decrease | -0.0697 |
| | The impulse response of SU to the environment | lnES→lnSU | Negative | Fluctuation-decrease | -0.0512 |
| | | lnEP→lnSU | Negative-positive | Increase-decrease | -0.0054 |
| | | lnER→lnSU | Positive | Decrease | 0.1275 |

**Note:** M1, M2, M3, and M4 represent the relationship models among PU, RU, EU, SU, and the environment. lnPU, lnRU, lnEU, and lnSU represent variables in the urbanization subsystem, whereas lnES, lnEP, and lnER represent variables in the environment subsystem.

positive impact on ecological pressure, while it has a negative impact on ecological status and response. This indicates that the increase and concentration of urban population will exert pressure on ecological status and response, but this pressure will gradually ease with the overall level of urbanization. It is worth noting that the increase and concentration of urban population will not increase ecological pressure, but rather alleviate it. This may be because as the population concentrates in cities, it is more conducive to the protection of the eco-environment, thus alleviating ecological pressure. From the perspective of the impact of ecological variables on urbanization, ecological response variables have a significant promoting effect on the increase and concentration of urban population, while ecological status variables show a fluctuating pattern. Ecological pressure variables have a certain negative constraint on urbanization, but the degree of constraint is limited.

(2) Impulse response between regional urbanization and the eco-environment(Fig 8C and 8D). Urbanization of space has positive, negative, and negative effects on ecological pressure, ecological status, and ecological response variables, respectively, with ecological status > ecological response > ecological pressure. This indicates that urban spatial expansion will to some extent alleviate eco-environmental pressure, but will significantly and strongly impose stress on ecological carrying capacity, while also increasing the process and difficulty of ecological response. Looking at the impact of eco-environmental variables on spatial urbanization, ecological pressure has an unclear effect on spatial urbanization, while ecological status

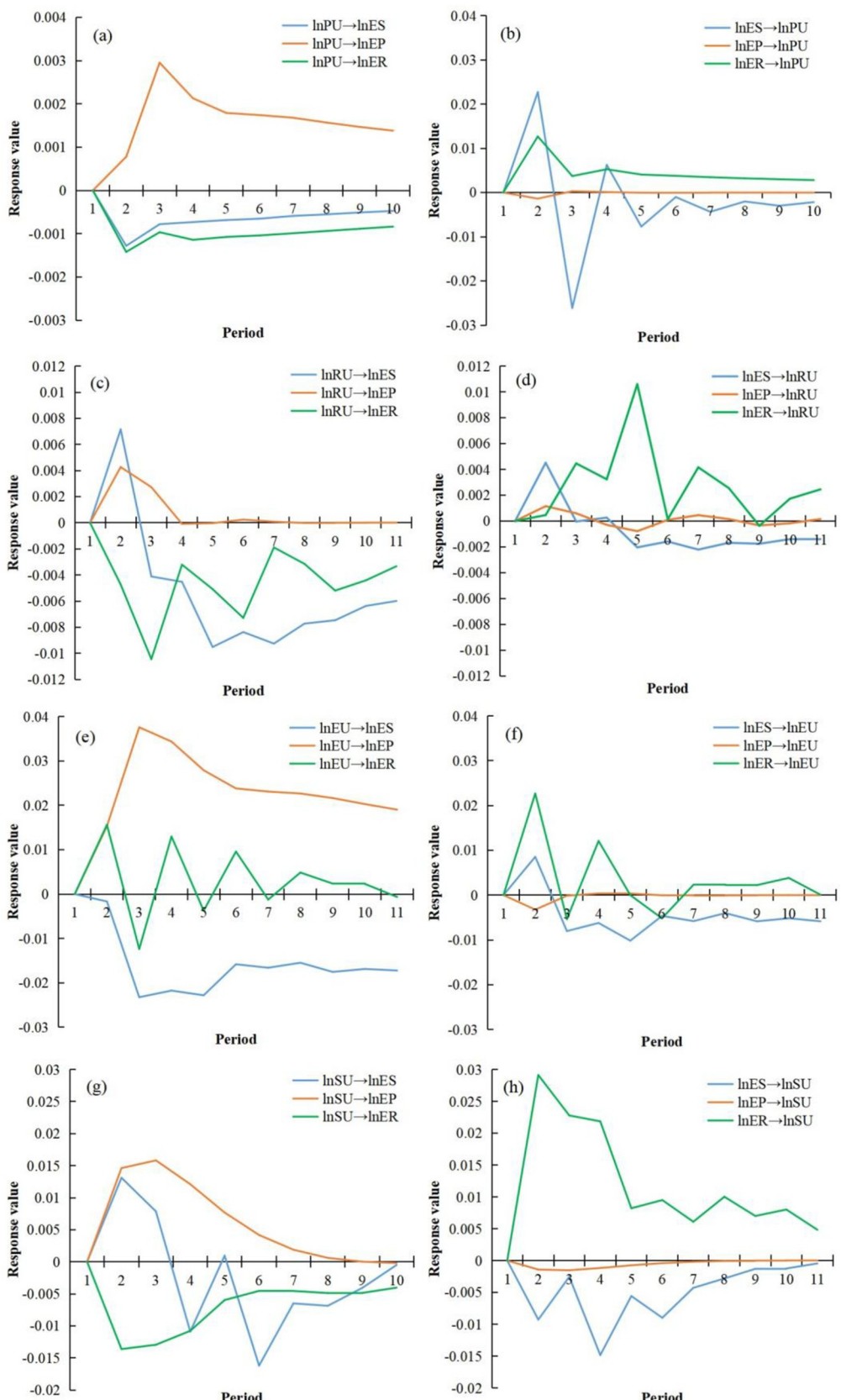

**Fig 8. Pulse responses of urbanization and the environment.** (a) Pulse response of environmental subsystem to population urbanization (PU). (b) Pulse response of the PU to the environment subsystem. (c) Pulse response of environmental subsystem to regional urbanization (RU). (d) Pulse response of RU to the environment subsystem. (e) Pulse response of the environmental subsystem to economic urbanization (EU). (f) Pulse response of economic urbanization to the EU subsystem. (g) Pulse response of environmental subsystem to social urbanization (SU). (h) Pulse response of SU to the environment subsystem.

has a significant inhibitory effect on urban spatial expansion. In addition, facing the impact of ecological response variables, spatial urbanization shows a violent positive fluctuation response process. This reflects that eco-environmental response measures have a promoting effect on urban spatial expansion, but this effect has intermittent characteristics. This may be closely related to the investment of environmental protection funds and urban pollution control.

(3) Impulse response between economic urbanization and environment(Fig 8E and 8F). In the face of the impact of economic urbanization, the response value of ecological pressure is all positive and has been maintained at a high level, indicating that economic development has a significant positive effect on the ecological pressure variable. This result implies that economic development does not increase eco-environmental pressure, but rather alleviates eco-environmental pressure. This deviates from our traditional understanding, and possible reasons are that economic development can improve production processes, reduce resource consumption, and increase investment to reduce the emission of "three wastes," thereby alleviating eco-environmental pressure. In addition, economic development can also impose significant stress on ecological conditions, while ecological response demonstrates a fluctuating process of both positive and negative effects. The impact of eco-environmental variables on economic urbanization is relatively weak. Ecological conditions can to some extent inhibit economic development, while the influence of ecological pressure on economic development is minimal. In the face of the impact of the ecological response subsystem, the response curve of economic urbanization fluctuates continuously in both positive and negative directions, reflecting the significant instability of this impact.

(4) Pulse response to social urbanization and the eco-environment(Fig 8G and 8H). From the perspective of the eco-environment's pulse response to social urbanization, social development and construction in the urbanization process have positively promoted the ecological pressure subsystem, while simultaneously constraining the ecological response process. This effect has a significant impact in the short term. Conversely, the ecological state subsystem has shown a clear negative effect on social urbanization, while the eco-environmental pressure on local social development and construction is not significant. In addition, in the face of the impact of the ecological response subsystem, social urbanization demonstrates a significant positive response, with an initial pronounced effect, followed by a gradual decrease in a "step-like" pattern. This changing process reflects the significant promotion of local social development in the short term as a result of the implementation of ecological response measures such as urban sewage treatment, waste disposal, industrial waste treatment, and environmental pollution control investment. However, this effect gradually weakens over time.

**4.3.3 Variance decomposition.** Fig 9 shows that the contribution rates of the eco-environment subsystems to urbanization development are relatively low, indicating that the feedback effect of the eco-environment on urbanization development is insignificant, which is consistent with the results of the pulse response analysis. In terms of feedback from the urbanization subsystems, the contribution rates of the environment subsystems to urbanization were ranked as follows: economic urbanization > social urbanization > regional urbanization > population urbanization.

The contribution rates of the urbanization subsystems to the changes in the eco-environment, as shown in Fig 10, were ranked as follows: ecological response > ecological

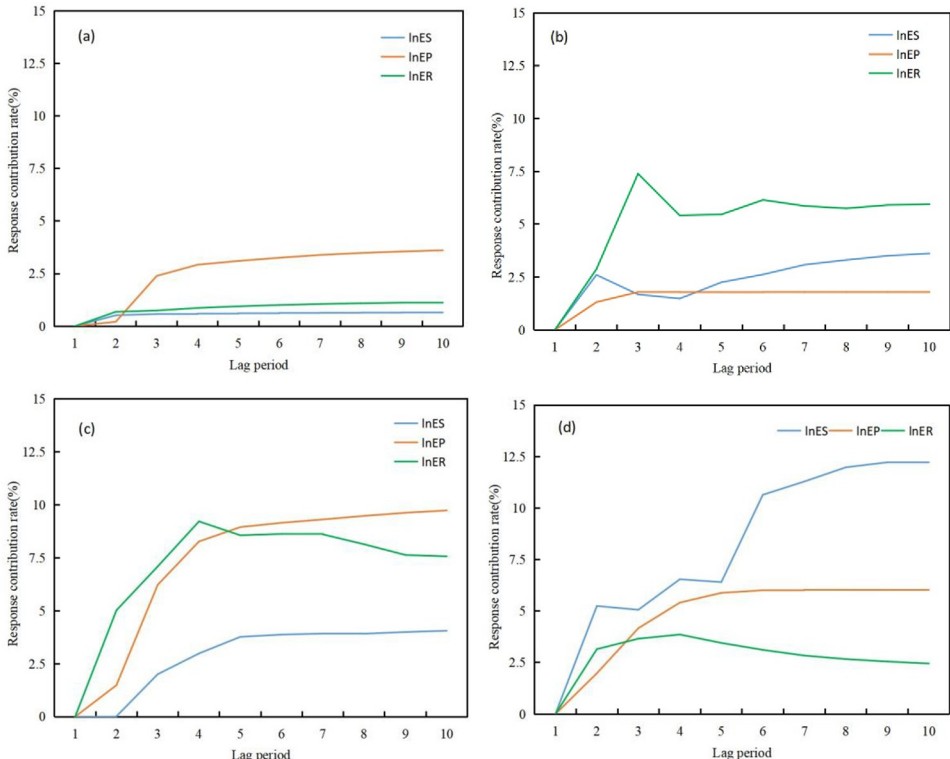

**Fig 9. Variance decomposition of the eco-environment on urbanization.** (a) Variance decomposition of the eco-environment of lnPUs. (b) Variance decomposition of the eco-environment on lnRU. (c) Variance decomposition of eco-environment in lnEUs. (d) Variance decomposition of the eco-environment on lnSU.

state > ecological pressure. First, the contribution rates of the urbanization subsystems to the ecological pressure subsystem are relatively low, further indicating that the pulse response of urbanization to ecological pressure is not significant and that ecological pressure has limited restrictive effects on local urbanization development. Second, in terms of the contribution rates of urbanization to the ecological state, population urbanization had the highest contribution rate, which was significantly higher than those of the other three subsystems, further confirming the significant pressure exerted by population growth and agglomeration on the ecological state. Finally, in terms of the contribution rates of urbanization to the ecological response, the contribution rates of economic and social urbanization were much higher than those of population and regional urbanization, indicating a strong demand for local economic and social development to improve the eco-environment in the region.

## 5. Discussion

### 5.1 Construction of the urbanization and eco-environment system evaluation index system

The construction of the index system is an important foundation for exploring the coordinated relationship between urbanization and the eco-environment. Urbanization is a complex dynamic process that involves changes in factors such as population, industry, society, space, and ecology. The eco-environment is a collection of interactions between resources, the environment, and organisms. In order to comprehensively reflect the urbanization process and the content of the eco-environment, people are gradually inclined to construct comprehensive

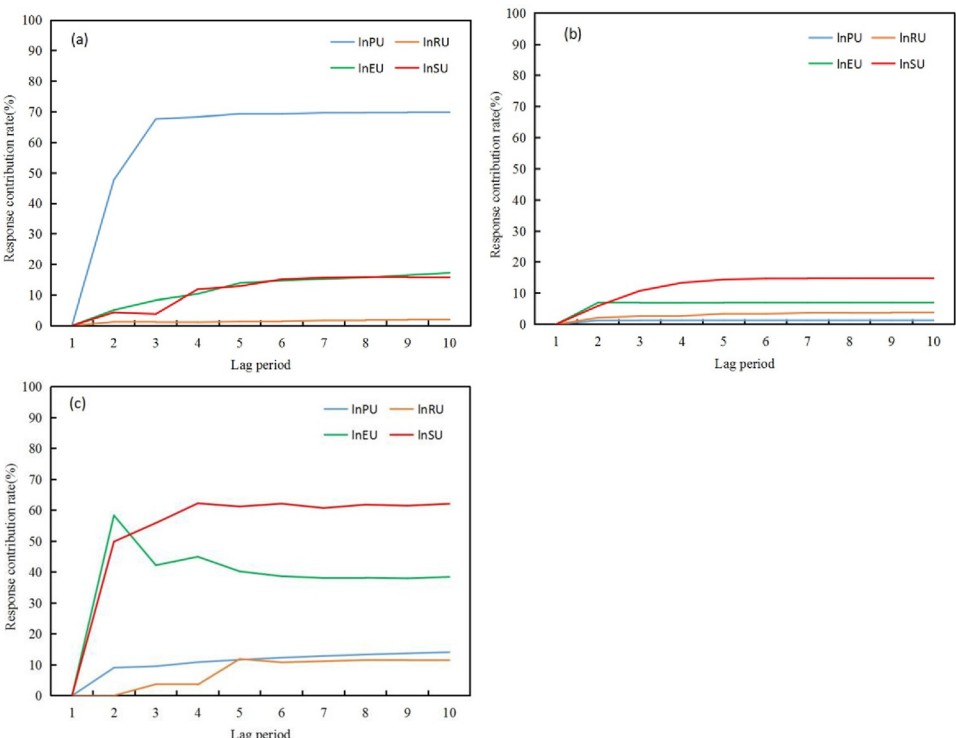

**Fig 10. Variance decomposition of urbanization on the eco-environment.** Variance decomposition of urbanization on lnES. (b) Variance decomposition of urbanization on lnEP. (c) Variance decomposition of urbanization on lnER.

index systems. Researchers usually choose reasonable evaluation indicators based on a "standard" framework, among which PESS and PSR are currently the most classic framework models for constructing urbanization and eco-environment index systems [43, 44]. Based on this, this paper constructs a comprehensive evaluation index system for urbanization system and eco-environment system. This section discusses and explains some of the indicators.

In terms of urbanization, the area of urban built-up areas is an important consideration. Some scholars directly use the total area of urban built-up areas as an indicator of spatial urbanization [45], while others choose per capita urban built-up area [46]. Since there are differences in the area of different research units, the total area of urban built-up areas cannot reflect these spatial size differences. At the same time, due to the regulations of urban planning standards, there is not much difference in per capita built-up area among cities, making it difficult to measure spatial urbanization. Urban area density is calculated by dividing the total area of urban built-up areas within the research unit by the total area of the research unit, which reflects the size of urban built-up areas per unit area and can more accurately reflect the urban spatial expansion of different-sized research units. Therefore, this paper selects urban area density as one of the indicators to measure spatial urbanization. Medical and health conditions are important aspects of social urbanization. In order to comprehensively reflect the medical and health levels of various cities, this article together selects the indicators of the number of medical technicians per 10,000 people and the number of beds in health institutions. In terms of eco-environment, many scholars have adopted air quality indicators. However, this article constructs an eco-environment system based on the PSR framework, which largely regards air quality as the comprehensive result of various indicators of ecological pressure, ecological conditions, and ecological responses. Therefore, air quality indicators are no longer selected

separately [47]. In addition, due to data limitations and difficulties in quantifying variables, some indicators have not been included, such as economic trade, education, and technology indicators that describe urbanization, as well as indicators that describe the eco-environment, such as soil and water pollution. In summary, based on previous research, considering the actual situation of the research area and the availability of data, this paper has carefully examined and compared, and comprehensively constructed an evaluation index system for urbanization and eco-environment. This index system will be further optimized in future studies of the same type.

## 5.2 High-level urbanization has a positive impact on the eco-environment

Analysis of the coupling relationships and coordinated development status revealed that cities with higher urbanization levels show higher levels of coupling and coordinated development between urbanization and the environment, even though these cities may have lagging environments. For example, the two major central cities of Chengdu and Chongqing have higher urbanization levels than other cities, and their environments are also lagging; however, the coupling and coordinated development between urbanization and the environment were the highest among all the cities in the cluster. This indicates that a high level of urbanization has a positive promotive effect on the coupling and coordinated development of urbanization with the environment. Cities with higher urbanization levels are more likely to develop in harmony with the environment. Additionally, the results of the VAR model analysis showed that urbanization has a positive promotive effect on the environment, particularly in high-level urbanization, where economic and social factors contribute to the feedback of the ecological response subsystem and thus help improve the environment. There are several possible explanations for this.

First, high-level urbanization is often accompanied by high-level economic development, which leads to greater financial investment in environmental protection and pollution control, thereby promoting environmental improvement. Additionally, urbanization drives technological innovation, and green technologies promote the production and use of green products, which contribute to environmental improvement. For example, the United Arab Emirates has reduced pollution caused by traditional energy industries by promoting clean energy industries [48]. Second, high-level urbanization implies a high level of social construction, which facilitates the implementation of environmentally friendly infrastructure and public services. Infrastructure such as piped water supply, sewage treatment, and waste disposal can be built, maintained, and operated more easily and economically in urban areas [49]. Third, urbanization has positive external dependencies and economies of scale that promote productivity improvement. Fewer resources are available to produce goods in urban areas, and highly urbanized cities have well-developed service industries that generate less pollution than traditional manufacturing development models. Therefore, urbanization is beneficial to the environment [42]. Fourth, the increase in urbanization has led to greater attention being paid to the environment and the pursuit of a clean and good living environment. This increased attention has promoted the implementation of environmental laws, government policies, and regulations to reduce environmental degradation [50].

## 5.3 Policy implications

Our research findings have important implications for policy making and promoting local development. (1) In terms of urbanization, different regions should adopt different urban development models to promote high-quality and balanced regional urbanization. The research results show that although the Coordination degree between urbanization and the

eco-environment is gradually increasing, most cities are still at a relatively low level. At the same time, there are significant spatial differences in Coordination degree, which is consistent with the law of urbanization development. Therefore, to improve the level of Coordination between urbanization and the eco-environment, the key lies in balancing the promotion of urbanization without harming the eco-environment. Different economic development and urbanization models should be adopted for cities in different regions with different resource endowments. For the two core cities of Chengdu and Chongqing, while strengthening their core status, their radiation effects should be enhanced to gradually promote the outward migration of labor-intensive industries and other non-core functions in an orderly manner, thereby driving the coordinated development of surrounding cities. For central cities in areas such as Mianyang, Yibin, Nanchong, and Wanzhou, interaction and cooperation with Chengdu and Chongqing should be strengthened to give full play to their regional radiation and driving effects, and to improve the overall functions and service levels of the cities. For mountainous cities surrounding urban clusters such as Ya'an, Dazhou, Kaizhou, Yunyang, and Qianjiang, infrastructure construction should be intensified to promote cooperation and circulation in transportation, logistics, information, and other aspects with regional central cities. At the same time, the development of characteristic advantageous industries should be strengthened to enhance urban vitality and attractiveness. (2) In terms of the eco-environment, efforts should be made to continuously optimize the eco-environment and enhance its carrying capacity. First, coordinated efforts should be made to jointly build and protect the regional eco-environment. Planning, construction, and protection of ecological space should be carried out together, and the ecological system of key ecological functional areas should be nurtured and restored. For cities with relatively low forest coverage in the central part of urban clusters, such as Ziyang, Neijiang, Zigong, Hechuan, and Rongchang, tree planting and afforestation should be strengthened to increase forest coverage and enhance the stability and resistance of the ecological system. For remote mountainous cities like Qianjiang, Yunyang, Ya'an, and Dazhou, environmental protection mechanisms and protection systems should be established, residents' environmental awareness should be enhanced, and the implementation of environmental protection work should be promoted effectively. Second, coordinated efforts should be made to address cross-border environmental pollution in the region. By formulating technical specifications and taking joint actions, pollution issues such as air, water, soil, hazardous waste, and noise should be collectively addressed.

## 5.4 Limitations and future directions

The Chengdu-Chongqing urban agglomeration is divided between two provincial-level administrative regions and consists of the "Chengdu Circle" and the "Chongqing Circle," representing a typical "dual-city model." Chengdu and the main urban area of Chongqing are the two core cities of the urban agglomeration, mutually attracting and influencing each other in the process of economic and social development. Other cities have also transcended provincial boundaries, with their populations, capital, cultures, and technologies integrating and intermingling with each other. In this process of attraction, influence, integration, and interconnectedness, the development of urbanization and the evolution of the eco-environment become more complex. Urban development not only affects the eco-environment within its administrative jurisdiction but also alters the eco-environment of neighboring cities or even more distant regions. Similarly, changes in the local eco-environment affect the development of urbanization in neighboring cities or distant regions. This phenomenon is referred to as the "telecoupling effect" between urbanization and the eco-environment, and a research framework for telecoupling has been proposed [51, 52] that consists of five basic components:

systems, flows, mediators, causes, and effects, which correspond to the questions regarding telecoupling that require further investigation in the future. This study may have some limitations in addressing these questions, which will be an important area for further research on the coupling effects and interactions between urbanization and the eco-environment in urban agglomerations.

## 6 Conclusions

Urban agglomerations are the areas where human activities and the eco-environment interact most intensively. In this study, we focused on the Chengdu-Chongqing urban agglomeration and analyzed the dynamic evolutionary relationship between urbanization and eco-environment development and explored their coupling relationship, coordinated development patterns, and interaction mechanisms. The findings of this study have important implications for sustainable development of urban agglomerations. The following are the conclusions of this study:

The Chengdu-Chongqing urban agglomeration has a relatively low level of urbanization and a good eco-environment background. From 2001 to 2020, both urbanization and the eco-environment quality gradually increased; however, the urbanization index increased more rapidly, resulting in increased pressure on the eco-environment. The evolution of urbanization and eco-environment indices shows significant spatial variations, with the formation of a high-value spatial agglomeration area of urbanization centered around Chengdu and Chongqing, known as the "dual-core" urbanization zone. Additionally, a low-value agglomeration area of urbanization exists in the northeastern part of the urban agglomeration, which encompasses cities such as Dazhou, Kaizhou, and Yunyang. The spatial pattern of the eco-environment also exhibited high values in the surrounding areas and low values in the central part of the urban agglomeration.

A close and stable coupling relationship exists between urbanization and the eco-environment, which has evolved into a high-level coupling stage. Urbanization and eco-environment systems demonstrate a good trend of coordinated development, gradually transitioning from a state of coexistence between disorder and low stability to a state of coexistence between low and moderate stability, and then to a state of coexistence between low, moderate, and high stability. Simultaneously, significant differences were observed in the state of coordinated development among cities, with multiple stable forms coexisting during the same period, indicating an imbalance in the development of cities within the Chengdu-Chongqing urban agglomeration.

Urbanization has a substantial impact on environmental changes, but no restrictive role of the eco-environment on urban development was observed. Various interactive relationships exist between urbanization and the eco-environment subsystems, including positive promotive and negative constraining effects. The positive promotive effect was mainly observed in the economic, social, and ecological response subsystems, while the negative constraining effect was mainly observed in the mutual coercion and inhibition between regional urbanization, economic urbanization, ecological status, and ecological pressure subsystems. Additionally, the intensity of the impact of each urbanization subsystem on the eco-environment was ranked as follows: economic urbanization > social urbanization > regional urbanization > population urbanization. The feedback effect of the eco-environment subsystem on urbanization was ranked according to ecological response, ecological status, and ecological pressure.

## Supporting information

**S1 Data.**
(XLSX)

**S2 Data.**
(XLSX)

## Acknowledgments

During the writing process of this article, we received assistance and support from Dr. Xiaoyu Gan from the School of Architecture and Environment at Sichuan University and Dr. Diwei Tang from the College of Forestry and Horticulture at Hubei Minzu University. We express our gratitude for their assistance. At the same time, We would like to thank Editage (www. editage.com) for English language editing.

## Author Contributions

**Conceptualization:** Weilong Wu, Bo Zhou.

**Data curation:** Yuzhou Zhang.

**Formal analysis:** Weilong Wu, Ying Huang.

**Funding acquisition:** Bo Zhou.

**Investigation:** Weilong Wu, Ying Huang, Yuzhou Zhang.

**Methodology:** Weilong Wu.

**Project administration:** Bo Zhou.

**Resources:** Bo Zhou.

**Software:** Yuzhou Zhang.

**Supervision:** Bo Zhou.

**Validation:** Weilong Wu, Ying Huang, Yuzhou Zhang.

**Visualization:** Weilong Wu.

**Writing – original draft:** Weilong Wu.

**Writing – review & editing:** Ying Huang.

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
