## [Decision Letter · Decision Letter 0]

13 Dec 2023

PONE-D-23-26168Research on the coupling relationship and interaction between urbanization and environment in city clusters: A case study of the Chengdu-Chongqing City ClustePLOS ONE

Dear Dr. Zhou,

Thank you for submitting your manuscript to PLOS ONE. After careful consideration, we feel that it has merit but does not fully meet PLOS ONE’s publication criteria as it currently stands. Therefore, we invite you to submit a revised version of the manuscript that addresses the points raised during the review process.

We look forward to receiving your revised manuscript.

Kind regards,

Xingwei Li, Ph.D.

Academic Editor

PLOS ONE

Journal Requirements:

This work was supported by the Key Research and Development Program of Sichuan Province (2023YFS0368).

5. Please amend either the title on the online submission form (via Edit Submission) or the title in the manuscript so that they are identical.

6. We notice that your supplementary figures are uploaded with the file type 'Figure'. Please amend the file type to 'Supporting Information'. Please ensure that each Supporting Information file has a legend listed in the manuscript after the references list.

Additional Editor Comments:

After peer review, the authors are invited to revise the manuscript according to their comments.

Reviewers' comments:

Reviewer's Responses to Questions

**Comments to the Author**

1. Is the manuscript technically sound, and do the data support the conclusions?

Reviewer #1: Yes

Reviewer #2: Yes

2. Has the statistical analysis been performed appropriately and rigorously? 

Reviewer #1: Yes

Reviewer #2: Yes

3. Have the authors made all data underlying the findings in their manuscript fully available?

Reviewer #1: Yes

Reviewer #2: Yes

4. Is the manuscript presented in an intelligible fashion and written in standard English?

Reviewer #1: Yes

Reviewer #2: Yes

5. Review Comments to the Author

Reviewer #1: The coordination relationship between urbanization and the environment is important for the regional sustainable economic and social development. The study has certain academic value, but I think there are still some problems that need to be revised:

1、The literature review does not provide a theoretical basis for the proposed method. There are at least four concepts of urbanization in this paper. It was not clearly specified to which this study refers.

2、Before conducting the empirical study, this paper did not carry out a theoretical analysis of the relationship between the environment and urbanization, and the path of interaction between them was not clear.

3、The indicator system needs further improvement. For example, there is an overlap in the meaning of SU5 and SU6, which all represent healthcare; and it lacks of air quality indicators in subsystem of ecological status, etc.

4、Result presentation is poor, as the results described in the text repeat information from the figures. Do not repeat the results. Be more creative, connect the results, point to the key information, provide some possible explanations for why the results show what they show,… The results as such are not the most important part of the research as they suffer from uncertainties. Instead the most important aspect is data discussion, interpretation (in understand the whole picture). The paper needs to be stronger in providing this.

5、Why authors chose 2001, 2008, 2014 and 2020 as typical year to analyze the spatial and temporal evolution patterns of urbanization and environment? Do these time points have a special meaning?

6、In section 3.1, authors need to provide a basis for grading urbanization and environmental levels.

7、line 406-408, the specific data is needed to support the results of the smoothness test and cointegration analysis.

8、In the coupled coordination analysis, the paper analyzed the level of coupled coordination from the perspective of the whole and individual cities, respectively, while in the impulse analysis, it only analyzed the impulse response between the subsystems of the whole urban agglomeration and lacked the analysis of the impulse response between the environment and urbanization of each city.

9、The figures of the paper needs further revision, such as figure 1 cannot reflect the location of city clusters in China, figure 2 and 3 need to change the location of the legend, compass, and scale.

Reviewer #2: 1. There are not enough innovations, and there are many related studies on the coupling and coordination and spatio-temporal analysis of urban agglomerations in the Chengdu-Chongqing area.

2. The format of the figure is not standardized.

3. The policy recommendations are vague and insufficient, with no substantive policy recommendations, and it is recommended to further supplement the detailed policy recommendations according to the quantitative regression results.

4. There is no Granger causality test in the VAR model, which can be considered.

5. The innovation points of this paper are not prominent, and it is recommended to clarify them at the end of the literature review.

6. The introduction part mentions the socio-ecological phenomenon of rapid urbanization, the relationship between urbanization and the environment, but the importance of coupling and interaction between them is insufficient, and it is suggested to add this part.

6. PLOS authors have the option to publish the peer review history of their article (what does this mean?). If published, this will include your full peer review and any attached files.

Reviewer #1: No

Reviewer #2: No

---

## [Author Response · Author response to Decision Letter 0]

1 Mar 2024

We have responded to the comments of the three reviewers one by one, please refer to "Response to Reviewers" for details.

---

## [Decision Letter · Decision Letter 1]

14 Mar 2024

Research on the coupling relationship and interaction between urbanization and eco-environment in Urban agglomerations: A case study of the Chengdu-Chongqing Urban agglomeration

PONE-D-23-26168R1

Dear Dr. Zhou,

We’re pleased to inform you that your manuscript has been judged scientifically suitable for publication and will be formally accepted for publication once it meets all outstanding technical requirements.

Kind regards,

Xingwei Li, Ph.D.

Academic Editor

PLOS ONE

Additional Editor Comments (optional):

Reviewers' comments:

Reviewer's Responses to Questions

**Comments to the Author**

1. If the authors have adequately addressed your comments raised in a previous round of review and you feel that this manuscript is now acceptable for publication, you may indicate that here to bypass the “Comments to the Author” section, enter your conflict of interest statement in the “Confidential to Editor” section, and submit your "Accept" recommendation.

Reviewer #1: All comments have been addressed

Reviewer #2: All comments have been addressed

2. Is the manuscript technically sound, and do the data support the conclusions?

Reviewer #1: Yes

Reviewer #2: Yes

3. Has the statistical analysis been performed appropriately and rigorously? 

Reviewer #1: Yes

Reviewer #2: Yes

4. Have the authors made all data underlying the findings in their manuscript fully available?

Reviewer #1: Yes

Reviewer #2: Yes

5. Is the manuscript presented in an intelligible fashion and written in standard English?

Reviewer #1: Yes

Reviewer #2: Yes

6. Review Comments to the Author

Reviewer #1: (No Response)

Reviewer #2: Most of my questions have been answered, which is a good job, but some details need to be paid more attention, such as the layout of the table.

7. PLOS authors have the option to publish the peer review history of their article (what does this mean?). If published, this will include your full peer review and any attached files.

Reviewer #1: No

Reviewer #2: No

---

## [Editor Report · Acceptance letter]

20 Mar 2024

PONE-D-23-26168R1 

PLOS ONE

Dear Dr. Zhou, 

I'm pleased to inform you that your manuscript has been deemed suitable for publication in PLOS ONE. Congratulations! Your manuscript is now being handed over to our production team.

Kind regards, 

on behalf of

Prof. Dr. Xingwei Li 

Academic Editor

PLOS ONE